# A sampler for atmospheric volatile organic compounds by copter unmanned aerial vehicles

Karena A. McKinney[1,2]*, Daniel Wang[2], Jianhuai Ye[2], Jean-Baptiste de Fouchier[2], Patricia C. Guimarães[3,4], Carla E. Batista[3,4], Rodrigo A. F. Souza[3,4], Eliane G. Alves[3,5], Dasa Gu[6], Alex B. Guenther[6], Scot T. Martin[2,7]*

[1]Department of Chemistry, Colby College, Waterville, Maine, 04901, USA

[2]School of Engineering and Applied Sciences, Harvard University, Cambridge, Massachusetts, 02138, USA

[3]Post-graduate Program in Climate and Environment, National Institute of Amazonia Research and Amazonas State University, Manaus, Amazonas, 69060-001, Brazil

[4]School of Technology, Amazonas State University, 69065-020, Manaus, Amazonas, Brazil

[5]Department of Biogeochemical Processes, Max Planck Institute for Biogeochemistry, Jena, Germany

[6]Department of Earth System Science, University of California, Irvine, California, 92697, USA

[7]Department of Earth and Planetary Sciences, Harvard University, Cambridge, Massachusetts, 02138, USA

Correspondence to: Karena A. McKinney (kamckinn@colby.edu) and Scot T. Martin (scot_martin@harvard.edu)

Submitted to *Atmospheric Measurement Techniques*

**Abstract.** A sampler for volatile organic compounds (VOCs) was developed for deployment on
a mulitcopter unmanned aerial vehicle (UAV). The sampler was designed to collect VOCs on up
to five commercially available VOC-adsorbent cartridges for subsequent offline analysis by
thermal-desorption gas chromatography. The sampler had a mass of 0.90 kg and dimensions of
19 cm $\times$ 20 cm $\times$ 5 cm. Power consumption was <3 Wh in a typical 30 min flight, representing
<3% of the total UAV battery capacity. Autonomous sampler operation and data collection in
flight were accomplished with a microcontroller. Sampling flows of 100 to 400 sccm were
possible, and a typical flow of 150 sccm was used to balance VOC capture efficiency with
sample volume. The overall minimum detection limit for the sampling volumes and the
analytical method was 3 ppt and the uncertainty the greater of 3 ppt or 20% for isoprene and
monoterpenes. The sampler was mounted to a commercially available UAV and flown in August
2017 over tropical forest in central Amazonia. Samples were collected sequentially for 10 min
each at several different altitude-latitude-longitude collection points. The species identified, their
concentrations, their uncertainties, and the possible effects of the UAV platform on the results
are presented and discussed in the context of the sampler design and capabilities. Finally, design
challenges and possibilities for next-generation samplers are addressed.

## 1. Introduction

Biogenic volatile organic compound (VOC) emissions from forests vary widely across plant species, ecosystem type, season, time of day, and environmental conditions at many scales, including from 10's to 100's of m (Gu et al., 2017;Fuentes et al., 2000;Goldstein and Galbally, 2007;Alves et al., 2018;Greenberg et al., 2004;Guenther et al., 2006;Klinger et al., 1998;Kuhn et al., 2004;Pugh et al., 2011;Wang et al., 2011). These variations can have significant effects on and be affected by atmospheric chemistry, air quality, and climate (Chameides et al., 1988;Fuentes et al., 2000;Laothawornkitkul et al., 2009;Goldstein et al., 2009;Kesselmeier et al., 2013;Peñuelas and Staudt, 2010). They may also be indicators of ecosystem change, plant health, and stress (Karl et al., 2008;Kravitz et al., 2016;Niinemets, 2010;Peñuelas and Llusià, 2003). Most field observations of biogenic VOC emissions are made from fixed-location towers, from tethered balloons, or from aircraft flying at high velocities well above the forest canopy (see Table 1 of Alves et al., (2016) for a summary of studies in the Amazon). As such, detailed information on the spatial distribution of emissions at 10's to 100's of meters has been difficult to obtain. This information is most critically needed in globally important and highly spatially heterogeneous source regions of VOCs, such as the Amazon, which is not well characterized even at large spatial scales. Thus, this scale is not represented in current VOC data sets, yet it is critical for understanding and quantitatively modeling VOC emission and uptake and is vital to advancing our present-day understanding of VOCs in atmospheric chemistry. New VOC measurements with increased horizontal coverage and resolution that could be used to test and improve existing emission models would be extremely valuable. Similarly, knowledge of VOC concentrations as a function of altitude throughout the boundary layer over a range of underlying

land cover types is needed to better constrain emissions, chemical reactions, and atmospheric
mixing of these compounds and to thereby inform atmospheric chemistry model development.

New approaches that are suited to spatially resolved sampling at these intermediate scales

is therefore needed by the atmospheric chemistry community.

Small, commercially available unmanned aerial vehicles (UAVs, commonly called

drones) have the potential to fill this gap in knowledge due to their extreme maneuverability
(Villa et al., 2016a). UAVs are available as either fixed wing aircraft, helicopters, or
multicopters. Multicopters (most often quad- or hexacopters) offer the advantages of being
highly maneuverable and easy to fly, as well as offering straightforward accessory mounting
options. Flight durations of up to 45 min and payload capacities of 6 kg are attainable with mid-
priced, commercially available copter-type UAVs. Development or adaptation of air sensors for
UAV platforms is, however, still in the early stages. To date, several researchers have utilized
UAVs to carry sensors to measure atmospheric trace gases in situ (Villa et al., (2016a) and
references therein.) Commercially available sensors for some trace gases (e.g., $CO_2$, CO, and
$NO_x$) are sufficiently compact to be carried by a UAV, but these are often limited by insufficient
sensitivity or difficult calibration (Cross et al., 2017). *In situ* techniques for quantifying VOCs at
the required sensitivity (< 10 ppt) are, however, large and complex instruments that exceed the
payload capacity of mid-range UAVs available to most researchers (Lindinger et al., 1998;Millet
et al., 2005;Blake et al., 2009;Kim et al., 2013).

As an alternative, the UAV platform offers the possibility to collect air samples for later

laboratory analysis. Black et al. (2018) used a commercial quadcopter to collect samples of
airborne mercury by drawing air through gold-coated quartz cartridges for later analysis by cold
vapor atomic fluorescence spectroscopy. The results showed the ability to resolve vertical
concentration profiles above a source and to differentiate between urban and rural mercury
concentrations. Although remote control of the sampler was not implemented, the authors
suggested this as a possible future improvement. Chang et al. (2016) demonstrated the use of a
whole air sampling apparatus mounted on a multicopter UAV platform to collect air samples for
off-line analysis. The sampler consisted of a single evacuated 2-L canister with a remote-
controlled valve actuated by a separate remote control unit independent of the UAV controller.
The flow rate and total sample volume was not monitored during flight. The authors successfully
detected VOCs, CO, $CO_2$, and $CH_4$ in the collected air samples and were able to distinguish
between samples collected upwind and downwind of an exhaust shaft. Both studies cite
maneuverability in three dimensions, spatial resolution, and the ability to evaluate emissions
from otherwise inaccessible locations as key advantages of UAV-based atmospheric sampling.
They also point out flight stability, an easily accessed and symmetrically positioned mounting
location, low cost, and lack of engine exhaust as features of battery-powered multicopters that
make them particularly well suited for environmental applications. As with any new sampling
method, the possible introduction of artifacts due to the platform should be considered. For the
case of UAVs, as with manned aircraft, the platform itself disturbs the surrounding air, which
could lead to issues such as loss of target species on surfaces, outgassing of interfering species,
or artifacts in measured concentrations due to enhanced mixing of the sample air. Nonetheless,
while the ability to detect atmospheric trace species and to map spatial gradients depends
strongly upon the target species, including its atmospheric variability and the detection threshold
of the analytical method, these several studies suggest that UAV-based sample collection is a
viable approach that promises to greatly expand access to previously inaccessible locations and
to provide a means to map spatial patterns in atmospheric trace species concentrations.
The use of VOC-adsorbent cartridges to capture VOCs from air with subsequent analysis
by thermal-desorption gas-chromatography mass spectrometry (TD-GC-MS) is well established
(Woolfenden, 2010a;Pankow et al., 2012). The adsorbent cartridges are small glass or metal
tubes, typically 9 cm in length and 0.64 cm in diameter. The cartridges are filled with a sorbent
material with a high affinity for VOCs. Woolfenden (2010b, a) and Pankow (2012) review the
performance of adsorbent cartridges for quantitative VOC measurements and compare their
retention and recovery of VOCs with whole air samples. Although whole air canisters have the
advantage of a very short (seconds) fill time, they are large (1 L volume) and heavy. Adsorbent
cartridge samples require longer sampling times, but their small size and light weight (10 g)
make them well suited to carrying on a UAV. The cartridges provide a lightweight, simple,
sensitive, and quantitative approach for determining a wide range of VOCs at ambient
atmospheric levels. The challenge is to design and construct an automated sample collection
system for cartridges suited to deployment on a multicopter UAV.
The primary scientific requirement of the sampler is that the total mass of analyte
collected be greater than the detection limit of the analytical system for that compound. In the
case of a volatile organic compounds detected by GC-MS, the detection limit has typically been
ca. 10 pg. Commercial detectors are now available with detection limits of < 1 pg, including the
GC-ToF-MS used for this study (Hoker et al., 2015), implying an order of magnitude lower
detectable VOC mixing ratios. The method detection limit also depends on the background level
of VOC measured in field blanks, which is also ca. 10 pg VOC. This corresponds to a VOC
detection limit of less than 10 pptv for a sample volume of a few liters of air, which can be
collected in 5 to 15 min by typical flow rates through adsorbent cartridges (Pankow et al., 2012).
This suggests that detection of VOCs in cartridge samples collected within current multicopter
flight durations of ca. 30 min is feasible. Automated operation of the cartridge sampler,
controlled either algorithmically based on elapsed time or position, or remotely by sending
commands to the sampler during flight, is desirable. Furthermore, the mass and dimensions of
the sampler must fit within the payload capacity of available UAV platforms. Herein, the design,
operation, and field validation of a VOC sampler using adsorption/thermal desorption cartridges
on a mid-size multicopter UAV that meets these requirements is described, and an example data
set collected in central Amazonia including a discussion of uncertainties is presented. The
possible effects of the UAV platform on the surrounding air and thereby on the collected sample
are an important consideration which is explored by computational fluid dynamics simulations.
**2. Experimental**
**2.1. Flight platform**

The UAV platform was a DJI Matrice 600 Professional Grade (Figure 1), which is a

hexacopter design with onboard stabilization. With propeller arms extended, the UAV measured
1.668 m across by 0.759 m high. Without the sampler attached, it weighed 9.6 kg with its six
batteries installed (model TB48S; 130 Wh, 18 V). The maximum ascent rate was 5 m s$^{-1}$, and the
maximum horizontal speed was 18 m s$^{-1}$. It had GPS positioning and maintained two-way
communication with DJI programs developed for iPad and Android tablet systems. The
positioning accuracy was $\pm0.5$ m in the vertical and $\pm1.5$ m in the horizontal. The maximum
flight time specified by the manufacturer was 40 min without a payload and 18 min for the
maximum payload mass of 5.5 kg at sea level. The VOC sampler was mounted to a mounting
frame underneath the UAV platform (DJI Matrice 600 Series Z15 Gimbal Mounting Connector
kit). Testing for the sampler load of this study indicated 25 min of flight time with a margin of
security of an additional 5 min. Actual battery use in each flight depended on the flight plan and
strength of local winds during the flight. The UAV was tested to a horizontal flight distance of
1000 m and a height of 150 m. A ceiling of 500 m above local ground level is hard-wired into
the device by the manufacturer.
**2.2. Sampler description**

Figure 2 shows the full system schematic, including the pump system flow paths and the

major power and signal connections within the sampler casing. The sampler requires a pump to
draw air flow through the sorbent cartridge, flow and pressure sensors, a flow regulation valve,
and a cartridge selection manifold to allow for multiple samples, as well as electronics to provide
power, issue commands, and collect data from the sensors during flight. The adsorbent cartridges
are positioned at the inlet of the flow path to ensure that the sample air does not come in contact
with any flow path surfaces prior to sampling as it could lead to contamination or loss of
analytes. The overall system layout of the sampler is designed to fit a standalone, modular form
factor in order to simplify installation and troubleshooting as well as to maximize
electromechanical compatibility with multiple UAV platforms in the field. A table with a
complete list of the sampler components is provided in the Supplement.

*Casing.* The sampling system resides in a rectangular acrylic casing that can be opened

for easy access for repairs and software updates to the onboard microcontroller. The completed
sampler measures 19 cm $\times$ 20 cm $\times$ 5 cm. The casing remains closed and attached to the chassis
of the UAV platform for exchanging sorbent cartridges between flights. The sampler casing is
directly integrated to the underside of the UAV chassis and does not interfere with standard
flight operations, including the functionality of the Matrice 600's automatically retracting
landing legs. The total sampler mass is 0.90 kg. The flight time decreases approximately linearly
with increasing payload mass below 5 kg. Based on the relationship between payload mass and
flight time provided by the UAV manufacturer, the decrease in flight time for a 1-kg payload is
estimated as 3.4 min (DJI.com).
*Flow system.* Cartridge sampling requires a sample stream at a calibrated flow rate in
order to determine the volume captured over the sampling period. The sample flow is drawn
through the system by a Parker CTS Micro Diaphragm pump, which can pull between 100 and
600 sccm of flow in a compact form factor. The volumetric flow of the pump is a function of the
pressure drop across the inlet and outlet, and is controlled via a manually adjustable pinch valve
(Model 44560; US Plastic Corp.) at the output of the flow system. The pump is driven by a 5.0
VDC brush-sleeve bearing motor.
A mass flow sensor (Model D6F-P; Omron) was installed upstream of the pump to
provide a continuous analog voltage output signal corresponding to the mass flow at standard
temperature and pressure. The flow sensor supports a flow range of 0 to 1000 sccm and includes
a built-in cyclone dust segregation system, which diverts particulates from the sensor element.
The mass flow sensor was calibrated periodically against a reference standard in the lab. The
mass flow sensor is used to calculate the total moles of gas in each sample (c.f., Section 2.4). The
flow sensor also serves as an indicator of sampler malfunction due to factors such as valve
failure or obstruction of the flow by debris during flight.
*Pressure system*. An absolute pressure transducer (MX4100AP; NXP) is positioned
adjacent to the flow sensor in order to measure the pressure in the flow path. The measured
pressure is used as a diagnostic of proper operation of the flow system. The device operates
across a pressure range of 20 to $10^5$ kPa. It outputs an analog voltage signal recorded by the
microcontroller that can be converted to a pressure value using a function provided by the
manufacturer. Laboratory calibration of the pressure sensor is possible but was deemed
unnecessary due to its purely diagnostic function.
*Manifold.* Activation of each sample cartridge is achieved with a solenoid valve manifold
(Model 161T102; NResearch Inc.) consisting of five independently actuated two-way, normally-
closed solenoid valves. All five valves have a nominal orifice of 1.0 mm and share a common
output port. The manifold is controlled by a valve driver board (CoolDrive Model 161D5X24;
NResearch Inc.). Valve actuation requires 200 mA at 24 V. The board uses a holding voltage that
is one third of the actuation voltage and is automatically achieved within 100 ms of activating the
solenoid. The five solenoid valves are independently controlled using 5 V logic level signals.
*Control system.* Autonomous sampler operation and data collection in flight is
accomplished with an Arduino Uno microcontroller. The microcontroller coordinates the
activation and operation of the pump and valves using a pre-programmed algorithm based on
elapsed flight time and collects data from the sensors.
*Electrical system.* The sampling system is powered by the UAV batteries via the 18 VDC
power output of the Matrice 600. The UAV power supplies two voltage regulators which provide
5 VDC output for the pump, pressure and flow sensors, Arduino Uno, and valve driver board,
and  24 VDC output for the valve manifold. The system consumes 2.5 Wh of electricity during a
30-min flight (25 min of sample time), which is less than 2% of the total UAV battery capacity.
The remaining 98% of battery capacity is available for UAV flight operations. The use of a
separate onboard battery to power the sampler was considered; however, the extra power
capacity was more than offset by the effect of the weight of an additional battery on total
available flight time.
**2.3. Sampling methods**
Air samples are collected using cartridge tubes packed with Tenax TA and Carbograph 5TD
(Markes International, Inc. C2 -AXXX-5149). Tenax TA is a relatively weak sorbent that
collects components with volatility less than benzene (e.g., $>C_6$) including monoterpenes, $C_{10}$,
and sesquiterpenes, $C_{15}$, whereas Carbograph 5TD shows strong sorbate affinity and captures
low-molecular-weight VOCs with carbon number of $C_3$ to $C_8$ (Woolfenden, 2010a) including
isoprene, $C_5$. The combination of these sorbent materials enables sampling of VOCs with carbon
number from $C_3$ to $C_{30}$, covering the expected range of atmospheric compounds from biogenic
and anthropogenic sources (Goldstein and Galbally, 2007). Both of the sorbent materials are
hydrophobic and suitable for air sampling at high RH conditions. Prior to sampling, tubes are
preconditioned at 320 °C for 2 h, then at 4 h at 330 °C for 4 h, and are then capped using 0.25-
inch (6.35-mm) Swagelok fittings with PTFE ferrules and kept sealed until they are installed on
the sampler just prior to flight.
The sorbent cartridges are mounted at the sampler inlet to ensure that the sample gas that
passes through the cartridges has not contacted other surfaces in the flow system, thus preventing
potential analyte losses or contamination from the flow system components. The cartridges are
oriented in a vertical position for sampling since horizontal installation can cause "channeling"
to occur as a result of sorbent falling away from the walls of the cartridge (ASTM International,
2015). No particle or ozone filter was used upstream of the cartridges to prevent loss of analytes
on the filter surfaces. Although a particle filter could be useful in preventing debris from entering
the sampling system, filters can also adsorb and later desorb semi-volatile VOCs, possibly
introducing sampling artifacts (Zhao et al., 2013). As this was judged to be a greater drawback,
an inlet filter was omitted. As such, both gas- and aerosol-phase VOCs are sampled; the reported
concentrations represent the sum of these contributions. The presence of ozone in the sample
cartridges may contribute to oxidation of the most reactive VOCs between collection and
analysis. The use of an ozone filter may help to mitigate this effect. The effect of ozone filters on
the samples is therefore being evaluated in ongoing work.
The total sample volume depends upon the flow rate and sample collection time. Both of
these parameters are easily adjusted in the field between flights. The flow is adjusted using the
manual pinch valve downstream of the pump. The flight time is programmed in the flight
algorithm executed by the Arduino Uno microcontroller. A constant low volumetric flow rate is
required to allow for optimal sorbent-sorbate interaction and uptake onto the sorbent matrix. A
target flow rate of 150 sccm was defined to maximize both VOC capture efficiency and sample
volume (Woolfenden, 2010b;Markes International Ltd., 2014). Based on the relationship
between sample volume and minimum detection limit reported by past studies (Pankow et al.,
2012), a minimum sampling volume of 1.5 L per adsorbent cartridge collected, corresponding to
ca. 2.5 ppt VOC, is targeted. This results in 10 min of sampling time per cartridge. Two to three
cartridge samples of this volume can be collected in a single flight while also carrying out take-
off/landing and transits between sampling locations. The Arduino Uno microcontroller provides
the operational flexibility to obtain smaller or larger sample volumes by utilizing either more
tubes and shorter collection times or fewer tubes and longer collection times, respectively, during
a single flight.
Alongside the sampling, blanks are collected to examine sampling artifacts such as
passive diffusion of VOCs into the tube. For the blanks, a sorption cartridge is installed on the
UAV and uncapped, but the sampling valve is not opened during flight. After sample collection,
the sample tubes and blanks are capped using the Swagelok fittings with PTFE ferrules, and
stored at room temperature. The collected tubes are transported from Brazil to USA for
chromatographic analysis. Tubes were analyzed within 1 week after collection. Under proper
transport and storage, sample artifacts have been shown to be minimal (Pollmann et al., 2005).
**2.4. Analysis by thermal desorption gas chromatography mass spectrometry (TD-GC-MS)**

The cartridge tubes are mounted into a thermally desorbing autosampler (TD-100,

Markes International, Inc). The VOCs are pre-concentrated at 10 °C followed by injection into a
gas chromatograph (GC, model 7890B, Agilent Technologies, Inc) equipped with time-of-flight
mass spectrometer (Markes BenchTOF-SeV) and flame ionization detector (TD-GC-
FID/TOFMS) (Woolfenden and McClenny, 1999;ASTM International, 2015). Internal standards
tetramethylethylene and decahydronaphtalene are injected into each sample after collection and
prior to analysis. The system is calibrated daily with a commercial standard from Apel-Riemer
Environmental Inc. (c.f. Supplement). The external gas standard is prepared using a dynamic
dilution system and the effluent is added to sorbent cartridges under conditions similar to those
used for sampling. The calibration cartridges are then analyzed using the same thermal
desorption GC analysis method. Response factors for additional VOCs are determined using
liquid standards injected on the cartridges or using FID signals by effective carbon number
(Faiola et al., 2012).

The mixing ratio $X_{VOC}$ of VOCs is calculated from the measured mass of each compound

in the sample and the volumetric flow rate according to the following governing equation:

$X_{VOC}$ = moles VOC / moles air = $(m_{VOC}\, R\, T) / (M_{VOC}\, P\, Q\, \tau)$         (Eq. 1)

where $m_{VOC}$ is the mass of the VOC measured in the sample, $M_{VOC}$ is the molar mass, $R$ is the
gas constant, $T$ is the temperature, $P$ is the pressure, $Q$ is the volumetric flow rate, and $\tau$ is the
sampling time. The mass flow sensor reports the equivalent volume of gas flow per unit time at
standard temperature and pressure conditions (273 K and 1 atm). Inserting these constant values
in Eq. 1 and combining them with R gives:

$X_{VOC}$ = moles VOC / moles air = ($m_{VOC}$ × *22400 sccm/mol*) / ($M_{VOC} Q_{std} \tau$)       (Eq. 2)

where $Q_{std}$ specifies mass flow. Thus, the measured quantities used in calculating $X_{VOC}$ are the
mass of VOC in the sample $m_{VOC}$, the mass flow rate $Q_{std}$, and the sampling time $\tau$. In practice,
since the mass flow rate can vary over the sampling period (Figure 3), a time integral of the
measured mass flow rate is used.

The detection limit of the GC-TOFMS analysis for isoprene is 1 pg, which is 0.25 ppt for

a 1.5-L sample. The detection limit of the measurement is, however, limited by the uncertainty in
the background (blank), which ranges from ca. 10 to 380 pg for the compounds shown in Table
1, equivalent to 2.5 ppt or 5%, whichever is greater, for a 1.5-L sample, and by the uncertainty in
the in-flight flow rate measurement, which is 15%. Combining these factors, the overall
uncertainty in the measured mixing ratio is then the greater of 3 ppt or 20%. A comparison of the
chromatograms of samples and blanks collected by the sampler with those collected on the tower
does not indicate the presence of any artifacts in the sampler cartridges attributed to outgassing
of volatile compounds from the UAV.
**2.5. Computational fluid dynamics (CFD) simulation**

CFD simulations are carried out using SOLIDWORKS Flow Simulation (Ver. 2017

SP3.0) (Waltham, USA). Dimensions and an input geometric model of the UAV are obtained
from the DJI company (DJI Downloads). A box with the dimensions and location of the sampler
is added to the geometry file. The propellers are simulated by discs of the same diameter, and to
simulate a hovering UAV a downward velocity of 11 m s$^{-1}$ is imposed through each disc so that
the lift produced by the motors balanced the system weight. The domain size was 2.4 m in width
and 2.0 m in height, with the UAV centered horizontally and at 1.2 m vertically.  An adaptive
grid was used, such that the grid spacing is smaller where gradients are larger. Boundary
conditions include atmospheric pressure far from the UAV, which is set to 1 atm. As the actual
pressure during sampling may differ from this value, it is used only as a baseline for comparison.
The results are optimized by performing iterations until the pressure difference between the last
two iterations was within 2 Pa, which corresponds to a change in speed of 0.004 m s$^{-1}$.
Uncertainties in the CFD simulations could arise from the choice of domain size or grid
resolution, which were limited by available computational resources, or assumptions such as the
use of solid disks to model the rotors. In flight the legs are retracted to horizontal. The
simulations do not account for possible changes to the circulation patterns due to the retraction of
the landing gear, although this effect is expected to be minor relative to the volume of the
disturbance created by the drone (c.f., Section 3).
**3. Results and discussion**

Samples were collected on August 2, 2017 of the dry season in central Amazonia at the

Manaus Botanical Gardens ("MUSA") of the Adolfo Ducke Forest Reserve. It is a 10 km × 10
km area set aside since 1963 to the north of Manaus, Amazonas, Brazil, and it has served as a
study site for several thousand publications. Three major terra firme forest classifications
describe the forest, including valley, slope, and plateau forests (Ribeiro et al., 1994;Oliveira et
al., 2008). The tree canopy height is typically in the range of 25 to 30 m. The UAV equipped
with the sample collector was launched and recovered from a platform of 3.5 m × 3.5 m atop a
42-m tower (3.0032° S, 59.9397° W, 120 m above sea level). Samples were collected on the
UAV at point A (3.0030° S, 59.9333° W, 122 m above sea level; Figure S1). The collection
point was 711 m from the launch point. The UAV successfully flew to the sample location
repeatedly based on pre-programmed GPS coordinates. Three samples were collected in separate
flights at heights of 60 m, 75 m, and 100 m relative to the ground level at the tower location. A
sample flow rate of 150 sccm and duration of 10 min duration were used to collect a total sample
volume of 1.5 L. For comparison, VOC collections were performed concurrently atop the MUSA
Tower with a hand-held motorized pump (Model 210-1002, SKC). These samples were collected
using a volumetric flow rate of 200 $cm^3$ $min^{-1}$ and sampling time of 20 min for a total sample
volume of 2.0 L. Mixing ratios were calculated from Eq. 1 using a pressure of 1.00 atm and
temperature of 25 ˚C (measurements of temperature and pressure were unavailable).
Uncertainties in pressure of +/-10% and temperature of ±5 C (±2%) were used to estimate an
overall uncertainty of 23% for the tower samples.

Data from the sampler showing flow and pressure for the three in-flight samples are

shown in Figure 3. To conserve battery power, the pump is turned off between samples and no
data are recorded. The results show that each valve successfully activated. After the initial start
up, a uniform flow rate of 150 sccm and a pressure of 1 atm is maintained during each sampling
period. The measured flow rate is used to calculate the volume of each sample to account for
small variations in flow.

VOC mixing ratios determined from samples collected by the UAV sampler and from

atop the tower are presented in Table 1. The raw mass measurements for each sample and blank
cartridge are included in the Supplement (Table S2). The results all fall within the expected
range of concentrations (e.g., ca. <1 – 10 ppb for isoprene) for the near-canopy environment over
the Amazon rainforest based on previous observations (Alves et al., 2016;Harley et al., 2004).
VOC emissions depend on many conditions, including season, time of day, temperature, light
levels (i.e., cloudiness),  and forest composition, which can vary on spatial scales of 10's of
meters. Atmospheric concentrations are also affected by atmospheric turbulent mixing and
photochemistry. It is therefore difficult to make direct comparisons among the samples presented
in Table 1, which were all collected at different locations (tower vs. point A), altitudes, and
times. More samples with systematic vertical, horizontal, and temporal coverage and a modeling
framework incorporating emissions, atmospheric mixing, and chemistry are needed in order to
draw firm scientific conclusions about the implications of atmospheric variability across these
coordinates. Further analysis and scientific interpretation of these results and a larger data set are
the subject of separate forthcoming publications.

The possible effects of air circulation created by the UAV multicopter rotors on the

sampling was considered.  Specifically, there were two main questions to be addressed. The first
was to determine the time scale at which the air in the sampling region beneath the UAV is
flushed. If the flushing time scale is significantly less than the sampling time, then, rather than
being drawn from a stagnant pool, the sampled air can be taken as representative of the
surrounding air. The second was to determine the spatial scale of the disturbance created by the
rotors, in order to assess whether smoothing of concentration gradients by rotor-induced mixing
is likely to influence the measured values.  Unlike many real-time sensors, which have
integration times on the order of a second, cartridge samples were collected over relatively long
time periods (minutes). Over this time period, atmospheric mixing serves to average out gas
concentration gradients at fine spatial scales (< a few m). Gradients at this scale would therefore
not be resolved by cartridge samples, even when not collected from a UAV platform. If the
spatial scale of mixing induced by the UAV is smaller than that of the atmosphere itself over the
sampling period, the perturbation of fine spatial scale gradients by the UAV circulation will not
significantly affect the measured concentrations. Hence, the second critical question to be
addressed by the CFD simulations is whether the spatial scale of atmospheric mixing induced by
the UAV rotors is larger than the spatial scale of atmospheric mixing over the sampling period. If
it is not, then the mixing due to the UAV should have little effect on the cartridge samples.

As there are no published computational fluid dynamics (CFD) studies specifically of the

DJI Matrice 600, CFD simulations of the UAV were performed. As shown in Fig. 4a, the
pressure difference between the area underneath the sampling box and the area under the
propellers was calculated as <100 Pa, indicating that the effect of the UAV on the pressure in the
sampling region is minimal. Because the mass flow sensor inherently accounts for changes in
sample pressure and temperature, small deviations in the pressure of the sampling region should
not affect the measured total mass of air sampled or the resulting VOC mixing ratio. This result
also suggests that any possible effects of UAV pressure fields on a pressure sensitive sensor
mounted in this area would be small.

Figure 4b shows the calculated air velocity distribution around the UAV. The simulation

suggests that air experiences roughly laminar downward flow from above the propellers,
undergoes turbulent recirculation to the UAV sampling region, and then is ejected below the
UAV. The simulation shows that the air flushing time in the sample region is fast (i.e., several
seconds) compared to the timescale of VOC sampling (i.e., 5-10 min). The disturbance due to the
rotors extends approximately 5 m above and below the UAV. This is consistent with the CFD
study by (Ventura Diaz and Yoon, 2018), which suggested that for their smaller quadcopter (1.2
kg), the sample represented an air parcel extending approximately 1 m above the UAV. As
expected for a larger drone, the disturbed air volume derived from Figure 4 is significantly larger
than in their study. The flow patterns, however, are remarkably similar considering the
simplifying assumptions and lower grid resolution used in this study (cf. Section 2.5), lending
credence to the general flow features shown in Figure 4.  The magnitudes of the pressure
variations around the UAV (+/-100 Pa, or +/- 0.10%) correspond to speed variations of ca. +/-0.2
m s$^{-1}$ or ca. 2 to 25% of speeds of 1 to 12 m s$^{-1}$. A 25% increase of the calculated speeds would
suggest a similar increase in the spatial scale for the dissipation of the resulting disturbance.
Hence, we estimate a range for the mixing scale of +/-5 to 7 m. The simulations thus indicate that
the sampler performs representative sampling of ambient VOC concentrations averaged across
±5 to 7 meters around the UAV. For comparison, the spatial scale of atmospheric vertical mixing
over the sampling period (10 min) can be estimated from the relationship $\Delta z = \sqrt{2K\tau}$, where $K$
is the eddy diffusivity, $\tau$ is the time period, and $\Delta z$ is the vertical distance. Estimates of the eddy
diffusivity within 10 m above a forest canopy are in the range of approximately 2 to 15 m$^2$ s$^{-1}$
during the day, though the values are uncertain and vary with local meteorology and canopy
roughness (Bryan et al., 2012;Saylor, 2013;Freire et al., 2017). $K$ then generally increases with
altitude for several hundred meters above the canopy (Wyngaard and Brost, 1984;Saylor, 2013).
Using the canopy-top values as a lower limit on the eddy diffusivity at the UAV height results in
an estimated lower limit on the vertical mixing scale of ca. 50 to 150 m, substantially larger than
that due to the UAV. A manuscript treating atmospheric mixing above the forest canopy more
explicitly using a large eddy simulation (LES) method is currently underway. Nevertheless, this
estimate suggests that mixing due to the UAV is expected to exert minimal influence on the
measured VOC mixing ratios.

As noted above, the sampled air is drawn systematically from above the altitude of the

UAV. It is therefore expected that the sampled air represents an altitude slightly higher than the
flight altitude. Based on a mixing volume extending 5 - 7 m above the drone, a vertical bias of
ca. -3 m altitude is inferred.

Several other studies investigated the effects of a multicopter on air sampling and reached

similar conclusions. Roldan et al. (2015) simulated flow around a quadcopter and validated the

simulations with air velocity measurements. The results showed that air speeds were greatest

near the propellers and smallest near the center of the UAV. The optimal location for air sensors

was at the center of the vehicle. Further testing involved measurements of $CO_2$ concentrations

with an onboard sensor near a $CO_2$ source, with and without the propellers rotating. There were

small differences (<5%) in the measured $CO_2$ concentrations, supporting the conclusions of the

simulations. Similarly, Black et al. (2018) demonstrated that no difference was observed in the

measured atmospheric mercury concentrations using a copter-based sampler when the UAV was

powered as compared to when it was unpowered. Together with the results of the current

simulations, these studies suggest that valid measurements of many atmospheric gas

concentrations can be obtained from multicopter platforms.

There are both advantages and disadvantages to mounting the sampler either atop or

beneath the UAV. The advantages of top mounting include faster time response and potentially

higher spatial resolution due to laminar flow and less mixing. Some disadvantages are the

potential for bias in some measurements, such as of particles, due to sampling from laminar flow

rather than well mixed air, and the potential for more vertical bias due to the strong laminar

downwash of air above the UAV. In addition, the temperatures at the top surface of the UAV

have been observed to become extremely hot (ca. 40 ˚C), particularly during the dry season. This

is particularly problematic for collecting VOCs on adsorbent cartridges, as the sampling

efficiency may be reduced at elevated temperatures. On the other hand, the advantages to

mounting beneath the UAV are that the sampler is protected from direct sunlight and therefore

cooler. Also, the flow beneath the UAV is well mixed, which avoids flow effects such as a bias

towards large particles. Disadvantages, such as mixing of concentration gradients and decreased
time resolution, are most significant for sensors with fast time response. A study by Villa et al.
(2016b), however, explored the differences in measured concentrations of a suite of trace gases
from a point source when the sensors were mounted above, below, and in the horizontal plane of
a hexacopter UAV. Their results show similar dilution of the plume measured above and below
the UAV, suggesting that the air sampled on top of the drone does not necessarily experience
less mixing. A sample inlet mounted such that it extends horizontally outside of the rotor wash
was the least affected by the UAV flow fields and could be a good solution for fast sensors. The
presence of eddies underneath the drone is less of an issue for our application, where samples are
collected over a 10 minute period. Atmospheric mixing and temporal averaging will smooth out
mixing ratio gradients over this time period, so mixing by drone-induced eddies should have
little effect on the measurement. Since the disadvantage of overheating if the sampler is mounted
on top of the UAV potentially outweighs the disadvantage of sampling from the turbulent flow
underneath, the decision to mount the sampler beneath the UAV is a reasonable one for this
particular application.

One of the key constraints on VOC sample collection by UAVs is the flight duration.

Although the manufacturer specifies a maximum flight time of 40 min, when carrying the
sampler under tested flight conditions and factoring in a margin of safety, the maximum flight
duration is limited to 25 min. Because the volumetric flow rate is also constrained to <200 sccm
for the manufacturer-recommended operation of the cartridges to avoid breakthrough, the
maximum air volume that can be collected during a flight is 5.0 L. Equation 1 in conjunction
with the method detection limit of 10 pg suggests a minimum detectable atmospheric mixing
ratio of 1 ppt for this sample volume at standard temperature and pressure. This sensitivity is
sufficient for abundant primary emissions such as isoprene and monoterpenes, which can have
mixing ratios of $10^2$ to $10^4$ ppt in tropical forests (Yáñez-Serrano et al., 2018). It may not,
however, be sufficient for quantifying primary compounds in other ecosystems with low-
emitting flora species, such as forests at higher latitudes or other ecosystem types such as
grasslands. It may also not allow for the detection of species of lower concentrations such as
sesquiterpenes. Characterization of these compounds is needed to fully understand the reactive
chemistry and aerosol formation potential of VOCs in forest environments. Additional strategies
to be explored for these compounds include more-rapid flow through the cartridge for low-
volatility compounds for which breakthrough is less of a concern or parallel sampling with
several cartridges simultaneously followed by common desorption at the TD-GC/MS.

There is a trade-off between the number of samples collected per flight and the individual

sample volume. Collecting multiple samples in one flight necessitates smaller volumes for each
sample and thus higher detection limits. Subject to the overall flight time limitation, the design of
the sampler allows flexibility in the sample count and duration to best achieve the experimental
objectives. For each individual flight, scientific choices can be made whether to collect a single,
large volume sample to target less-abundant species or multiple smaller samples for surveying
the major VOC components.

A number of strategies can ameliorate these limitations. To facilitate the continuous

operation of the UAV, multiple sets of batteries can be used. One set is charged while another set
is in use. After each flight, the depleted batteries can be replaced with the spare fully charged set
and the UAV launched immediately instead of waiting for the batteries to charge. This allows the
number of samples collected to be maximized. Extension of the sample time can also be
achieved by initiating a sample on one flight, pausing while the UAV returns for battery
replacement, then returning to the same location and resuming collection with the same
cartridge. A modification on this approach would be to use a single cartridge to collect air at the
same location and time of day over multiple days, resulting in an average for that time period.

A major goal of ongoing development is to enable control of sampler functions and

collection of sampler data from the tablet-based UAV control software, either manually or as
part of a pre-programmed GPS-based flight algorithm. In the current version, the flight trajectory
is programmed with the UAV control software, whereas and sampler operation is controlled by a
stand-alone program on the Arduino Uno microcontroller. The two programs are synchronized in
time from initialization with a short time buffer so that the UAV arrives at the sampling location
1 min prior to opening the valve. In order to fully integrate these functions, real-time
communication among the sampler, the UAV on-board computer, and the user control interface
on the tablet is required. The Arduino Uno microcontroller does not have the capability to
communicate with the UAV on-board computer. To address this issue, the next step in the
development is the replacement the Arduino Uno microcontroller with a Raspberry Pi miniature
computer. Communication between the sampler and user interface also can enable development
of custom software as a diagnostic tool that enables monitoring the status of the valves and pump
during the flight. This capability can be important to alert the user to problems during flight, such
as the failure of valves or the pump to be activated, as has occurred occasionally on windy days
(5% of flights with winds $>4$ m s$^{-1}$) due to strong vibration. This failure mode has largely been
eliminated by reinforcing the electrical connections and inspecting them before each flight.

Current regulations in some locations, including the US, require that the operator

maintain visual contact with the UAV. This was also deemed best practice in the current study as
users gained experience and comfort with flight operations. Launching the UAV from a tower
permitted the pilot to maintain visual contact during flight. As another approach, the UAV
sampler has also been flown in locations with hills where it is possible to visualize the top of the
canopy over an area of lower elevation from an area of higher elevation. In the future, as
regulations permit, navigation from the ground to above the canopy should be possible and
would allow sampling in more remote and densely forested regions. A clearing of sufficient size
to allow the UAV to be navigated would be required. A camera to provide remote visualization,
either on the same drone or on a second companion drone, would aid in navigation outside of the
pilots visual range.
Together with the flight capabilities offered by modern day UAV platforms, this sampler
opens the door to studying VOC emission and uptake at previously inaccessible scales. In the
long term, data from this project will shed light on atmospheric chemistry, biodiversity, and
ecosystem stress within the context of global climate change.

**Acknowledgments.** Support from the Harvard Climate Change Solutions Fund is gratefully

acknowledged. The Museu de Amazonia of the Manaus Botanical Gardens kindly provided

access and logistical support. A Senior Visitor Research Grant of the Amazonas State Research

Foundation (FAPEAM) is acknowledged.

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

**Table 1.** Summary of biogenic VOC types and concentrations collected on 2 August 2017. Results are shown for sample collection by

the UAV-based sampler at 711 m from the tower launch location as well as by use of a hand-held pump at the top of the

tower. Local time is -4 h to UTC. The overall uncertainty is the greater of 3 ppt or 20% for the UAV samples and 3 ppt or

23% for the tower samples. [a]Samping height as relative to ground level at the MUSA tower. [b]Only major monoterpenes are

listed here. In addition to isoprene and monoterpenes, four sesquiterpenes including β-caryophyllene were detected. [c]"n.d."

denotes that the VOC concentration was below the detection limit of the instrument.

| Sample | Local time | Location (Distance to Tower, m) | Sampling height[a] (m) | Isoprene (ppt) | α-Pinene (ppt) | β-Pinene (ppt) | d-Limonene (ppt) | Tricyclene (ppt) | Athujene (ppt) | Camphene (ppt) | Carene (ppt) | Total monoterpene[b] (ppt) |
|---|---|---|---|---|---|---|---|---|---|---|---|---|
| 1 | 11:15 - 11:35 | 711 m | 75 | 1282.9 | 45.0 | 9.9 | 5.3 | 1.1 | 2.3 | 0.9 | n.d. | 78.8 |
| 2 | 11:15 - 11:35 | Tower top | 42 | 2017.8 | 93.4 | 18.0 | n.d.[c] | 0.7 | 5.1 | n.d. | n.d. | 118.3 |
| 3 | 13:15 - 13:35 | 711 m | 100 | 2672.9 | 55.0 | 12.6 | 10.5 | 0.8 | 2.5 | 0.7 | 0.4 | 94.1 |
| 4 | 15:15 - 15:35 | 711 m | 60 | 1724.1 | 49.2 | 11.4 | n.d. | 1.7 | 2.8 | 3.7 | 0.3 | 84.0 |
| 5 | 15:15 - 15:35 | Tower top | 42 | 2539.3 | 57.1 | 10.7 | 0.5 | 0.3 | 3.8 | 0.3 | 0.2 | 73.0 |

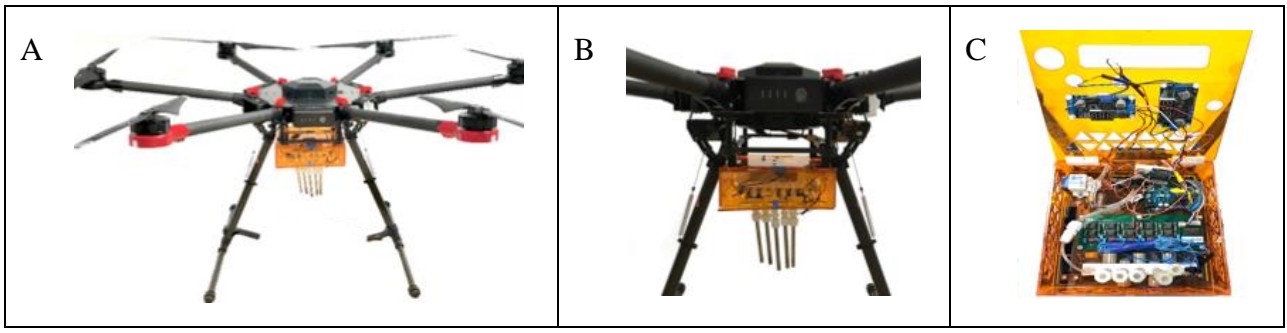

**Figure 1.** UAV equipped with VOC sampler: (A) DJI Matrice 600 hexacopter UAV. (B) Custom-built sampler visible in orange mounted to UAV. Five VOC sorbent cartridges (Markes International, Inc) are seen on the undercarriage. (C) Sampler with lid open to show pump and electronics package seen in panel B for differentially actuating sample flow through the sorbent cartridges.

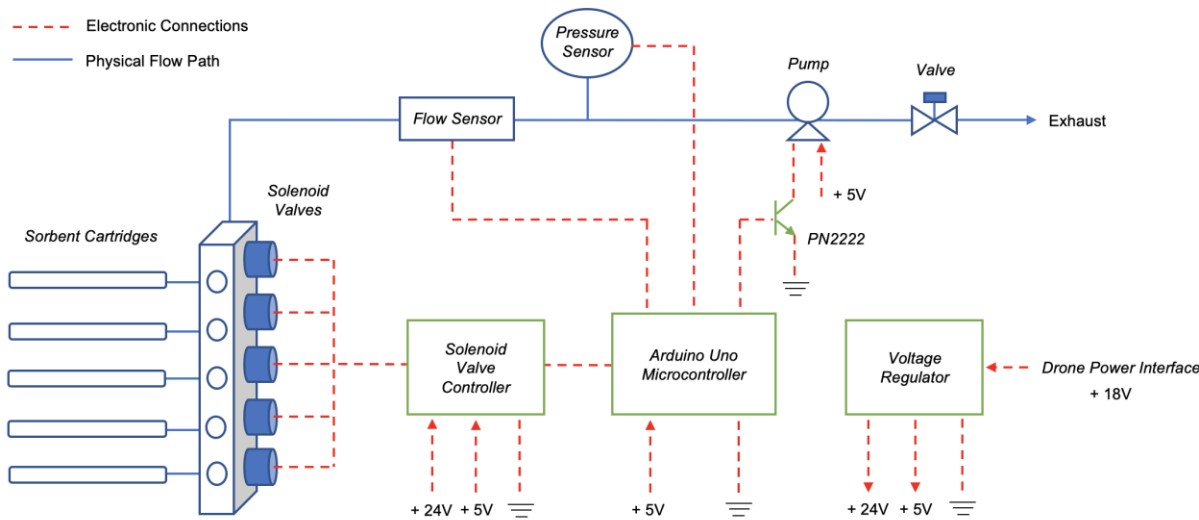

**Figure 2.** Schematic diagram of sampling device. All components are powered by the UAV
batteries through the 18 VDC power output on the Matrice 600 and are controlled by
an Arduino Uno microcontroller. Gas flows from the ambient atmosphere through the
sorbent cartridges and out to the pump and exhaust.

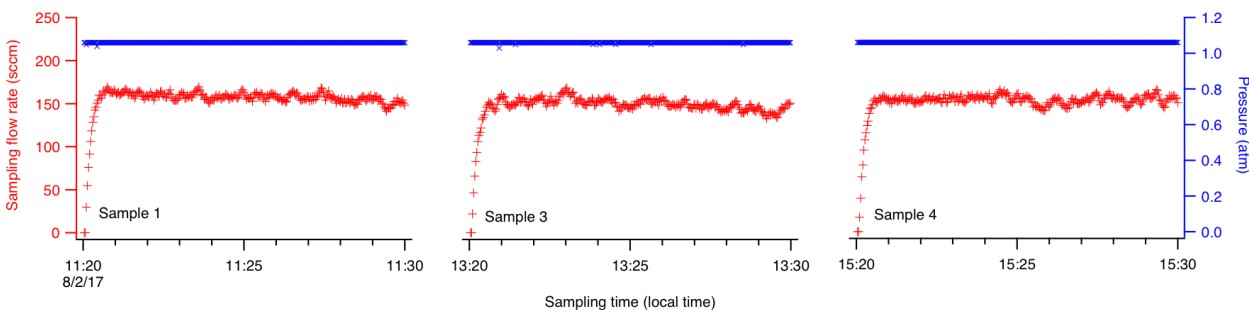

**Figure 3.** Time series of diagnostic data collected during the VOC-sampling UAV flights.

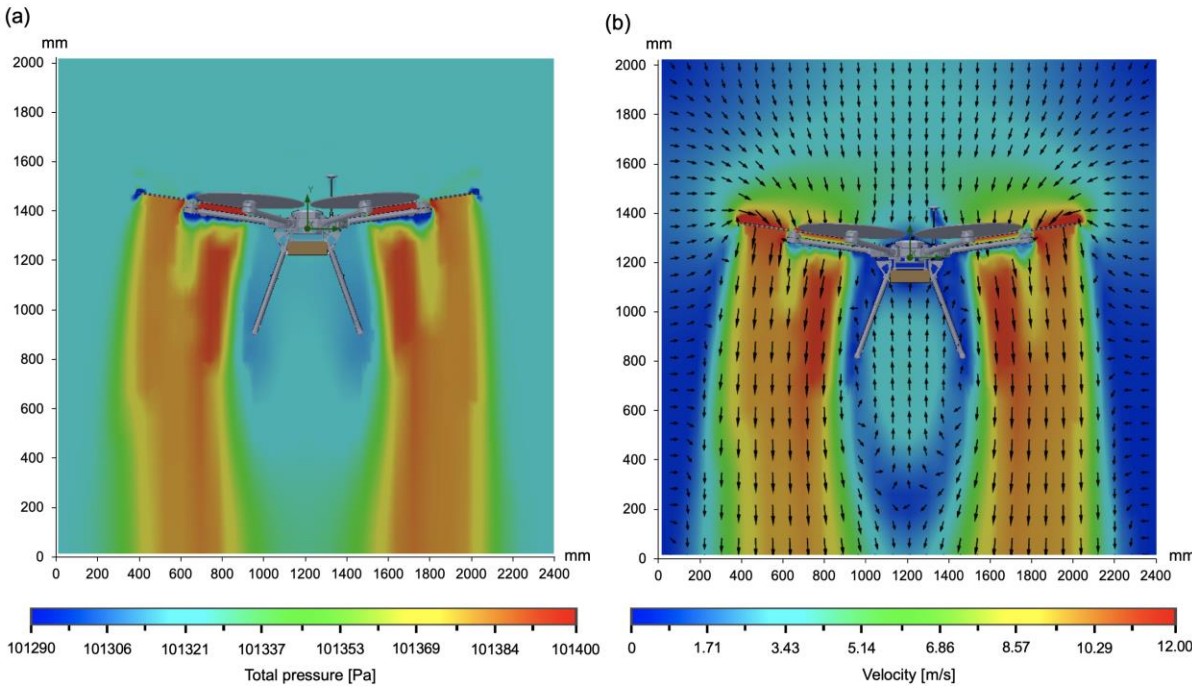

**Figure 4.** (a) Vertical pressure distribution and (b) air velocity distribution around the UAV from the CFD simulation. Pressure difference between the UAV sampling area and the area under the propellers was simulated to be less than 100 Pa indicating a minimal effect of pressure on sampling. The air velocity was 1.65 m s$^{-1}$ upward around UAV sampling region, suggesting a fast air flushing time underneath the sampling box.