# Peer review of "A sampler for atmospheric volatile organic compounds by copter unmanned aerial vehicles"

_Atmospheric Measurement Techniques, 2018_

## Referee Comment (RC1) · Anonymous Referee #1 · 26 Sep 2018

Overall, this is a well-written paper and a valuable technology. It should be published with minor revisions, however, there are some important discussion points and details that I would like to see addressed.

General Comments: 1) The dilution due to rotor-wash, which is a problem for all instruments without an inlet that extends beyond the turbulence induced by the multi-rotor platform, is not discussed until later in the paper. The authors conclude that their samples are representative of ambient mixing ratios; however, while this may be the case for isoprene and monoterpenes, the carbon fiber DJI M600 Pro body likely has some emissions of low molecular weight VOCs, which could pose problems for cartridge

measurements of other VOCs. Second, simulations are shown for the legs extending in the landing position, although I imagine samples were collected when the legs are retracted (as is done automatically by the M600 software after takeoff). The differences in the flow with legs retracted or if samples were collected when the legs were in the landing position should be discussed.

2) The challenges associated with desorption of VOCs and OVOCs from cartridges and quantitative measurements of these compounds compared with whole air samples should be discussed.

3) Please discuss how atmospheric temperature was measured. For instance, what sensor was used to measure temperature, and was this done in the flow path as well or elsewhere on the sampling platform? This appears to be a critical measurement for determining the mixing ratios of VOCs, and it is not explicitly described anywhere.

4) A comparison of samples and blanks would be very useful in demonstrating the utility of this platform.

Specific Comments:

Line 2: Word Choic. Why "copter technology," not "multi-rotor"?

Line 10: The phrase "close to 2 ppt" is vague. Please be more specific, and include the "3ppt or 20% (total) uncertainty in measured mixing ratios" in the abstract.

Line 27: delete "and" and insert comma and "from" before "tethered balloons"

Line 30-31: Which is less well characterized, horizontal gradients or vertical gradients at these scales? Discuss which of these is more important for models.

Line 31-35: "Thus, this scale . . .global atmosphere" Pease rewrite these sentences, as they read awkwardly. Also, what does "the primary scale for VOC emission" mean? Is that the finest resolution that models are able to represent? Also, "precisely the missing link" maybe be slightly overstating the importance of these measurements to

understanding of VOCs in atmospheric chemistry (i.e. we don't know if this is the "only" missing link, and indeed, it likely is not). Finally, if these measurements are scarcer in the amazon then elsewhere, cite some studies that have adequately captured this horizontal or vertical resolution in other parts of the world, and discuss how it has informed our understanding of regional emissions and the subsequent atmospheric chemistry.

Line 35: replace "height" with "altitude"

Line 55-77: Although there are a number of advantages to multirotor UAV platforms, it would be helpful to discuss the importance of rotor-wash and potential of sample dilution due to rotor-wash (see general comment 1). I see this is in part addressed later in the paper, however, this should also be mentioned in the introduction.

Line 92-92: Is the detection limit of the VOCs entirely determined by the subsequent analysis (e.g. GC-MS or GC-ToF-MS)?

Line 93-94: This sentence isn't needed and is vague (please delete): "this suggests that detection of VOCs from multicoptor flight..."

Line 95: insert "cartridge" prior to "sampler"

Line 115 and Figure 2: Label and discuss the 18V supply from the DJI M600 pro to the cartridge sampler, and its integration.

Line 139: Delete "the" before "cartridge sampling"

Line 152: Please comment in the text (here) on whether in the future, the use of filters prior to the cartridges could be helpful in preventing debris from making its way into the system. I see, filters are finally mentioned on Line 195, however, I think this should be discussed more fully and earlier.

Line 156: Please comment here on how atmospheric temperature was measured (see general comment 3)?

Line 157: "It outputs analog voltage. . ." Is the same is true of the mass flow sensor, as well (i.e. produce an analog voltage that is converted into a flow value? Also, is this conversion based on laboratory or manufacturer based calibrations? Please comment in the text.

Line 162: Please comment on the inline, wetted of solenoid valves and their potential VOC emissions to which cartridge samples could be exposed. Could this influence the detection limit of this system, particularly with sensitive analyzers such as GC-ToF-MS?

Line 170: Are there additional sensors to system pressure and system flow on the sampling platform? If not, please specifically list these two sensors.

Line 172: "via the power distribution board" is awkward phrasing- consider rewording.

L182- L189: Discuss the benefits of be able to measure high molecular weight compounds (C9-C30) of this approach, compared with others.

Line 204-207: Do you base your sample volume collection on prior measurements in different environments? Can this be adjusted easily in the field or between flights?

Line 213-215: "not influence the results"- can you expand on this?

Line 221-222: Are these internal standards injected prior to sample collection as well or simply prior to sample analysis? Please explain this in the text.

Line 240: This is a good description of the uncertainty and the detection limit. This detection limit and uncertainty do not seem compatible with the "nearly 2 ppt" listed in the abstract. Are they? If so, please explain.

Line 242: Please 1) discuss the purpose of the CFD simulations and 2) the uncertainties in the SOLIDWORKS Flow simulations.

Line 264: It would be worthwhile to discuss the influence of rotor-wash potentially on measurements and their differences at altitudes of 60 m, 75 m, and 100 m. Are these measurements representative of 60 +/- 5m? Also note if these samples were taken on

ascending vertical profiles or separate flights (related to general comment 1).

Line 267: Were cartridges at the tower collected using an identical cartridge sampling system, including a pressure sensor in the flow path and a mass flow sensor or only a pump? Please describe this in the text.

Line 285-290: Discuss in the text more explicitly what the impact is of deviations in pressure in the sampling region. How would this specifically impact the representativeness of cartridge measurements?

L346-347: This second half of this sentence is a bit confusing. Isn't pre-programed GPS-based operation already employed? Is the goal to integrate that seamlessly into the DJI flight software? L356: How high were the winds on these days that operation of the solenoid, pump or sensors failed? How typical are winds this high?

P22 (Figure 4): The M600 Pro is not typically flown (and I imagine samples aren't collected) with the legs down for landing. How is the flow in these simulations altered when the M600 legs are retracted, if at all? See general comment 1.

P22 (Figure 4): Please add a vertical scale and horizontal scale on Fig. 4a and Fig. 4b.

Line 264: It would be worthwhile to discuss the influence of rotor-wash potentially on measurements and their differences at altitudes of 60 m, 75 m, and 100 m. Are these measurements representative of 60 +/- 5m? Also note if these samples were taken on ascending vertical profiles or separate flights (related to general comment 1).

Line 267: Were cartridges at the tower collected using an identical cartridge sampling system, including a pressure sensor in the flow path and a mass flow sensor or only a pump? Please describe this in the text.

Line 285-290 : Discuss in the text more explicitly what the impact is of deviations in pressure in the sampling region. How would this specifically impact the representativeness of cartridge measurements?

L346-347: This second half of this sentence is a bit confusing. Isn't pre-programed GPS-based operation already employed? Is the goal to integrate that seamlessly into the DJI flight software?

L356: How high were the winds on these days that operation of the solenoid, pump or sensors failed? How typical are winds this high?

P22 (Figure 4): The M600 Pro is not typically flown (and I imagine samples aren't collected) with the legs down for landing. How is the flow in these simulations altered when the M600 legs are retracted, if at all? See general comment 1.

P22 (Figure 4): Please add a vertical scale and horizontal scale on Fig. 4a and Fig. 4b.

---

## Referee Comment (RC2) · Anonymous Referee #2 · 3 Dec 2018

General Comments This is a very well-written manuscript describing the development of a VOC sampler for autonomous, drone-based sampling. The motivation and relevant background is thoroughly but succinctly presented in the introduction, and the platform and results are clearly and generally well-described. I recommend publication of the manuscript, pending the authors: 1) add some context for what results should be expected for vertical distribution of VOCs in Table 1, so that the reader can better interpret the results presented here, and 2) more satisfactorily explore the vertical sampling bias introduced by rotors drawing air down from above (or gather comments from an additional reviewer with substantial experience with the fluid dynamics of drones). The CFD analysis is laudable, but does not conform to experience in working with large drones

with payloads, where vertical disruption of plumes extends greater than 5 m in many cases, and the paper cited to suggest < 1 m disruption is based on drone platforms that are substantially smaller.

Specific comments 111 – Noteworthy that the sampler was placed on the platform underneath the drone. Downwash and eddies present a significant challenge in sampling underneath drones (as you explore later), leading many to mount sensors on top of the drone, where flow is laminar, or to extend a sampling inlet outside the rotor influence. CFD simulations are a helpful place to start, but ultimately you can learn a lot by just flying your specific platform through a smoke plume. You'll notice straight, laminar flow lines on top that extend from several meters above (depending on system mass) and a mess of eddies underneath. Dave Barrett and Scott Hersey at Olin College of Engineering presented on this in collaboration with Aerodyne at AAAR and AGU in 2016 – check their materials for more clues. This eddy issue matters less for your application than for their 1-Hz instrument, since you are not after time-dependent (i.e. highly spatially resolved) data, but rather bulk VOC mass over an entire flight segment. But is nonetheless an important consideration. Explore options to mount on top, or to extend a sampling inlet to a point horizontally outside rotor influence.

240 – CFD simulation parameters are described, though it's not explicit at this point why you did CFD simulation (I can assume where you're headed). I suggest giving some sense of the need/purpose for this simulation before introducing it.

258 – The drone was launched and recovered from a platform above the canopy, but one of the key motivations for the drone-based sampling platform is to avoid the need for platforms and to be able to access more remote sampling locations. Can you speak to the usability of this platform in the types of contexts that motivate the study (i.e. those with dense canopies and no platforms)?

262 – Given the note above, and the high velocity of air flow down through the rotors of the drone, I am not convinced that 60 m actually represented 60 m. I should be clear

that I see your exploration of this with CFD modeling, but your model results conflict with my experience seeing drones sample smoke plumes in the field. With a slightly larger drone (S900) and slightly heavier payload (2.5 kg), I consistently see rotors draw down air from several (>/= 5) meters above mounted instruments in buoyant plumes. Experience suggests to me that your vertical sampling bias is greater than the 1 m suggested in line 294. Further, the result suggesting 1 m vertical bias in air sampling based on rotor air flow in Diaz and Yoon (2018) is based on a significantly smaller drone with no payload. Your large drone with payload will, necessarily, exert a greater vertical impact on air flows than theirs. This comment comes with the caveat that I am basing them solely on experience and observations with quad copters, and no modeling or detailed analysis of my own. I recommend either a brief review of this section – especially as it relates to altitude-of-sample bias – by a reviewer with greater expertise in the fluid mechanics of multi-rotor aircraft, or an addition of language that outlines the potential for vertical sampling bias on the order of several meters.

278 – "Reasonable consistency" is subjective. Quantify, and compare with either sampling+measurement uncertainties or previously published variability in VOC concentrations with height above canopy (or both).

282 – CFD modeling appears. I applaud the authors for attempting to address rotor influence in sampling. Ultimately, as I stated above, I expect the below-drone air flow perturbations to be less important for your application of 10 min resolution samples. But the bias introduced in the vertical resolution is of concern and my experience tells me that for a drone your size, the vertical extent of air disruption is substantially greater than the 1 m suggested here, based on results from a much smaller drone platform with no payload. I am, unfortunately, not the right reviewer to critique your CFD model run, and suggest that an additional reviewer explore this.

Table 1 – Can you put these results in context that help the reader understand the consistency of measurements and how they conform to expectation? For example, I notice that isoprene concentrations vary substantially with altitude, though not in a way

that decays with altitude (as I might expect). Same with Pinene(s). As presented, I'm unable to discern why the 100 m sample at the sampling site has higher concentrations of monoterpenes than both the 60 m and 75 m sample. Can anything be determined from ratios of VOCs to tell what's going on here? What should I expect to see in vertical variability? This doesn't conform to my expectations of reducing concentration with altitude, so please explore this so that the reader isn't left with questions about whether sampling bias or the drone platform is responsible.

---

## Author Comment (AC1) · 30 Jan 2019

**Response to Reviews: A sampler for atmospheric volatile organic compounds** by copter unmanned aerial vehicles**

Karena A. McKinney, Daniel Wang, Jianhuai Ye, Jean-Baptiste de Fouchier, Patricia C. Guimarães, Carla E. Batista, Rodrigo A. F. Souza, Eliane G. Alves, Dasa Gu, Alex B. Guenther, and Scot T. Martin

Reviewer comments are in **bold**, the authors' response is in Plain text, and revisions to the manuscript are in *Blue italics*. Please note that for consistency, all line numbers refer to the version of the manuscript published in the Discussion forum.

**Reviewer 1**

**Overall, this is a well-written paper and a valuable technology. It should be published with minor revisions, however, there are some important discussion points and details that I would like to see addressed.**

We thank the reviewer for the extremely helpful and detailed comments, which have led to significant improvements to the manuscript. Our responses to the comments and corresponding changes to the manuscript are described below.

**General Comments: 1) The dilution due to rotor-wash, which is a problem for all instruments without an inlet that extends beyond the turbulence induced by the multi-rotor platform, is not discussed until later in the paper.**

The issue of influence of rotor-induced turbulence and the need for CFD simulations to understand its effects have been introduced earlier in the paper. Specifically, we have made the following additions to the Abstract and Introduction:

Abstract: "The species identified, their concentrations, their uncertainties, and the possible effects of the UAV platform on the results are presented and discussed in the context of the sampler design and capabilities."

Introduction, line 71: "As with any new sampling method, the possible introduction of artifacts due to the platform should be considered. For the case of UAVs, as with manned aircraft, the platform itself disturbs the surrounding air, which could lead to issues such as loss of target species on surfaces, outgassing of interfering species, or artifacts in measured concentrations due to enhanced mixing of the sample air."

Introduction, line 100: "The possible effects of the UAV platform on the surrounding air and thereby on the collected sample are an important consideration which is explored by computational fluid dynamics simulations."

The authors conclude that their samples are representative of ambient mixing ratios; however, while this may be the case for isoprene and monoterpenes, the carbon fiber DJI

**M600 Pro body likely has some emissions of low molecular weight VOCs, which could pose problems for cartridge measurements of other VOCs.**

Any emissions from the carbon fiber of the drone would be diluted in the ambient air flow past the drone, which is large due to the motion of the rotors. We therefore expect that any resulting interferences would be small. It is nonetheless worth investigating. One way of assessing such artifacts would be the blank cartridges obtained during flight. We have added to the Supplement a table of VOC masses detected in the cartridge samples and blanks collected onboard the UAV and from the tower. A comparison between the blanks obtained in flight and on the tower shows similar background levels, suggesting that outgassing from the drone does not interfere with the targeted VOCs. We have added the following statement to Section 2.4:

*Line 239: "A comparison of the chromatograms of samples and blanks collected by the sampler with those collected on the tower does not indicate the presence of any artifacts in the sampler cartridges attributed to outgassing of volatile compounds from the drone."*

Second, simulations are shown for the legs extending in the landing position, although I imagine samples were collected when the legs are retracted (as is done automatically by the M600 software after takeoff). The differences in the flow with legs retracted or if samples were collected when the legs were in the landing position should be discussed.

In order to put it in the context of other changes, we respond to this comment in more detail below. Please refer to the comment on P22 (Figure 4).

**2) The challenges associated with desorption of VOCs and OVOCs from cartridges and quantitative measurements of these compounds compared with whole air samples should be discussed.**

There are numerous other studies of cartridge sampling that have addressed this issue. We consider it outside the scope of this study to include a full review here. Instead, we have added a few sentences to the text referring the reader to some of the key publications on this topic and comparing the pros and cons of adsorbent cartridges vs whole air samples.

"Woolfenden (2010b, a) and Pankow (2012) review the performance of adsorbent cartridges for quantitative VOC measurements and compare their retention and recovery of VOCs with whole air samples. Although whole air canisters have the advantage of a very short (second) fill time, they are large (1 L volume) and heavy. Adsorbent cartridge samples require longer sampling times, but their small size and light weight make them well suited to carrying on a UAV."

**3) Please discuss how atmospheric temperature was measured. For instance, what sensor was used to measure temperature, and was this done in the flow path as well or elsewhere on the sampling platform? This appears to be a critical measurement for determining the mixing ratios of VOCs, and it is not explicitly described anywhere.**

We thank the reviewer for this comment, which drew our attention to an aspect of the measurement description that was unclear. Temperature was not measured by the sampler, but because the flow sensor measures mass flow, not volume flow, temperature is not needed to

calculate moles of sampled air or VOC mixing ratio (nor is pressure). Mass flow sensors operate by measuring the dissipation of heat by the gas flow. This quantity depends on the mass (or moles) of gas passing the sensor element per unit time, so it inherently accounts for changes in temperature and pressure. Mass flow is reported in standard cm3 min-1 (sccm), which can be converted to moles of gas per minute using the molar volume of an ideal gas at standard conditions (273 K and 1 atm). Hence, measured ambient temperature and pressure are not used in the calculation. We have revised the text to remove references to calculating the volume flow rate and to clarify how the mixing ratio calculation was performed. The revisions are shown below.

Line 149: "*The mass flow rate is converted into a volumetric flow rate using the measured pressure at the flow sensor and atmospheric temperature. The sample volume is obtained by integrating the volumetric flow rate over time.* The mass flow sensor is used to calculate the total moles of gas in each sample (c.f., Section 2.4)."

Line 154: "The measured pressure is also used with atmospheric temperature to convert mass flow rate to volumetric flow rate as UAV altitude changes used as a diagnostic of proper operation of the flow system."

Line 228: "The mixing ratio Xvoc of VOCs is calculated from the measured mass of each compound in the sample and the volumetric flow rate according to the following governing equation:

$$Xvoc = moles VOC / moles air = (mvoc R T) / (Mvoc P Q \tau)$$
(Eq. 1)

where mvoc is the mass of the VOC measured in the sample, Mvoc is the molar mass, R is the gas constant, T is the temperature, P is the pressure, Q is the volumetric flow rate, and  $\tau$  is the sampling time. The mass flow sensor reports the equivalent volume of gas flow per unit time at standard temperature and pressure conditions (273 K and 1 atm). Inserting these constant values in Eq. 1 and combining them with R gives:

$$X_{VOC} = moles \ VOC \ / \ moles \ air = (m_{VOC} \times 22400 \ sccm/mol) \ / \ (M_{VOC} \ Q_{std} \ \tau)$$
(Eq. 2)

where  $Q_{std}$  specifies mass flow. Thus, the measured quantities used in calculating Xvoc are the mass of VOC in the sample  $m_{VOC}$ , the mass flow rate  $Q_{std}$ , and the sampling time  $\tau$ . In practice, since the mass flow rate can vary over the sampling period (Figure 3), a time integral of the measured mass flow rate is used."

**4) A comparison of samples and blanks would be very useful in demonstrating the utility of this platform.**

A table of measured VOC masses in the samples and blanks has been added to the Supplement (Table S2) and has been referenced in the text. The table shows that for isoprene and  $\alpha$ - and  $\beta$ -pinene, the mass of VOC in the samples is well in excess of that in the blanks. We have also updated the data in both Table 1 and Table S2 based on a re-analysis of the original GC data. In doing so, we noted that in the original manuscript the mixing ratio values in Table 1 were based on a preliminary analysis rather than the final calibration data. As a result of applying the final

calibration data, the mixing ratios have changed substantially, in some cases by a factor of 2. The revised values are accurate to within the stated uncertainties.

**Specific Comments:**

**Line 2: Word Choice. Why "copter technology," not "multi-rotor"?**

We have taken the reviewer's suggestion, but have changed the wording to "multicopter" rather than "multi-rotor" as multicopter is consistent with the terminology that is used in the title and throughout the manuscript.

**Line 10: The phrase "close to 2 ppt" is vague. Please be more specific, and include the "3ppt or 20% (total) uncertainty in measured mixing ratios" in the abstract.**

The sentence in the abstract has been revised to read: "*The overall minimum detection limit for the sampling volumes and the analytical method was 3 ppt and the uncertainty the greater of 3 ppt or 20% for isoprene and monoterpenes.*"

**Line 27: delete "and" and insert comma and "from" before "tethered balloons"**

The suggested revision has been made.

**Line 30-31: Which is less well characterized, horizontal gradients or vertical gradients at these scales? Discuss which of these is more important for models.**

Neither is particularly well characterized. Most measurements are made from towers, so most represent a single point both horizontally and vertically (some tall towers have multiple points in the vertical). A single tower observation is often assumed to be representative of a large geographical area or land cover type. The extent to which this is true has not been fully investigated and can depend on the region, with the tropics exhibiting greater horizontal heterogeneity than temperate forests. Emission models are 2-dimensional (i.e., land surface only). The most widely used of these, MEGAN, has a horizontal resolution of 1 km. The resolution is based on land cover data and emissions are calculated based on the distribution of plant types at each grid point. (Emissions are not directly interpolated from tower measurements, though these measurements can be used to validate the model.) Thus, it would likely be straightforward to use measurements with higher horizontal resolution to test and improve existing emission models, which would be an important advance. On the other hand, regional or global models do not resolve near-canopy vertical gradients in VOCs. This mainly done only in a small number of (generally 1-D) canopy-scale models that have been used in isolated studies. Vertical gradients may therefore not impact models as directly, but the results are important for understanding the interplay of mixing, deposition, and chemical processes in determining the fate of VOCs, and therefore for informing model development more generally. We have added comments on this topic to the text. They are shown combined with revisions in response to the next comment, below.

Line 31-35: "Thus, this scale . . .global atmosphere" Pease rewrite these sentences, as they read awkwardly. Also, what does "the primary scale for VOC emission" mean? Is that the finest resolution that models are able to represent? Also, "precisely the missing link"

maybe be slightly overstating the importance of these measurements to understanding of VOCs in atmospheric chemistry (i.e. we don't know if this is the "only" missing link, and indeed, it likely is not). Finally, if these measurements are scarcer in the amazon then elsewhere, cite some studies that have adequately captured this horizontal or vertical resolution in other parts of the world, and discuss how it has informed our understanding of regional emissions and the subsequent atmospheric chemistry.

As suggested by the reviewer, we have rewritten these sentences. Individual VOC measurement sites are scarcer in the Amazon than elsewhere, but we know of no existing data sets anywhere that capture the horizontal heterogeneity of forest emissions with a resolution of 10's to 100's of meters. We have clarified this point in the revised text:

"As such, detailed information on the spatial distribution of emissions at 10's to 100's of meters has been difficult to obtain. This information is most critically needed in globally important and highly spatially heterogeneous source regions of VOCs, such as the Amazon, which is not well characterized even at large spatial scales. Thus, this scale is not represented in current VOC data sets, yet it reflects the primary scale is critical for understanding and quantitatively modeling VOC emission and uptake and is precisely the missing link in vital to advancing our present-day understanding of VOCs in atmospheric chemistry. This information is even more scarce in remote areas, such as the Amazon rainforest, that are very important sources of VOCs to the global atmosphere. New VOC measurements with increased horizontal coverage and resolution that could be used to test and improve existing emission models would be extremely valuable. In addition-Similarly, knowledge of VOC concentrations as a function of altitude height throughout the boundary layer over a range of underlying land cover types is needed to better constrain emissions, chemical reactions, and atmospheric mixing of these compounds and to thereby inform atmospheric chemistry model development."

**Line 35: replace "height" with "altitude"**

The suggested revision has been made.

Line 55-77: Although there are a number of advantages to multirotor UAV platforms, it would be helpful to discuss the importance of rotor-wash and potential of sample dilution due to rotor-wash (see general comment 1). I see this is in part addressed later in the paper, however, this should also be mentioned in the introduction.

As discussed in response to General Comment 1 above, we have added language addressing the importance of rotor-induced mixing for sampling and motivating the CFD simulations in the introduction. We have also expanded the discussion of the implications of the CFD simulations in the Results and Discussion section. These changes are described in more detail below, in response to the comment on Line 262.

**Line 92-92: Is the detection limit of the VOCs entirely determined by the subsequent analysis (e.g. GC-MS or GC-ToF-MS)?**

No, the detection limit is determined by the detection limit of the analysis method, combined with the background levels measured for the field blanks. The uncertainty in the measurement

also depends on the uncertainty in the measured flow rate. These factors are detailed in Section 2.4. This section of the Introduction is intended as a demonstration that drone-based cartridge sampling is feasible, not as a detailed discussion of uncertainties. We have, however, added a sentence regarding the role of the measured VOC background level in determining the detection limit to the text, as follows:

"The primary scientific requirement of the sampler is that the total mass of analyte collected be greater than the detection limit of the analytical system for that compound. In the case of a volatile organic compounds detected by GC-MS, the detection limit has typically been ca. 10 pg. For a sample volume of a few liters of air, which can be collected in 5 to 15 min by typical flow rates through adsorbent cartridges, this corresponds to a VOC detection limit of less than 10 pptv (Pankow et al., 2012). Commercial detectors are now available with detection limits of < 1 pg, including the GC-ToF-MS used for this study (Hoker et al., 2015), implying an order of magnitude lower detectable VOC mixing ratios. The method detection limit also depends on the background level of VOC measured in field blanks, which is also ca. 10 pg VOC. This corresponds to a VOC detection limit of a few liters of air, which can be collected in 5 to 15 min by typical flow rates through adsorbent cartridges (Pankow et al., 2012)."

**Line 93-94: This sentence isn't needed and is vague (please delete): "this suggests that detection of VOCs from multicoptor flight. . ."**

This sentence is the conclusion of the preceding exercise in determining the required sampling time and demonstrating that drone-based adsorbent cartridge sampling is feasible. We think it is an important point, so we have chosen to retain it. To reduce vagueness, we have specified the drone flight duration to which we are referring. The revised text reads as follows:

"This suggests that detection of VOCs in cartridge samples collected within current multicopter flight durations of ca. 30 min is feasible."

**Line 95: insert "cartridge" prior to "sampler"**

The suggested revision has been made.

**Line 115 and Figure 2: Label and discuss the 18V supply from the DJI M600 pro to the cartridge sampler, and its integration.**

The figure has been revised to more clearly label the 18V power supply from the UAV. The relevant section of the figure caption has been revised as follows:

**"All components are powered by onboard batteries on the UAV batteries through the 18 VDC power output on the Matrice 600 and are controlled by an Arduino Uno microcontroller."**

The section of the text referenced here (Line 115) is intended to be a description of the drone platform itself. The electrical interface to the sampler is discussed in a later part of Section 2.4 labeled *Electrical System*. To address the reviewer's comment, we have revised that section to more clearly describe the electrical interface between the drone and the sampler. Please see the response to the comment on Line 172 below for the revised text.

**Line 139: Delete "the" before "cartridge sampling"**

The suggested revision has been made.

**Line 152: Please comment in the text (here) on whether in the future, the use of filters prior to the cartridges could be helpful in preventing debris from making its way into the system. I see, filters are finally mentioned on Line 195, however, I think this should be discussed more fully and earlier.**

There is a statement in the text that the flow sensor can be used as an indicator of a malfunction such as blockage of the flow by debris. In practice, this issue has not arisen during use of the sampler. We have elected not to use a filter on the inlet since filters can adsorb and later desorb semi-volatile VOCs, leading to artifacts. On balance, the disadvantage of potential filter artifacts outweighs the benefit of using a filter to prevent the low-probability chance of obstruction. We have added several sentences, shown below, explaining this reasoning to the existing text. The discussion of filters appears in Section 2.3 Sampling Methods. After careful consideration, we have elected not to move this material earlier in the manuscript. Line 152 is in Section 2.2, which is a description of the sampler design and operation. The use of filters is more germane to the discussion of sample handling in Section 2.3. Moving it earlier would, we believe, lead to less clarity between these two topics.

"No particle or ozone filter was used upstream of the cartridges to prevent loss of analytes on the filter surfaces. Although an inlet filter could be useful in preventing debris from entering the sampling system, filters can also adsorb and later desorb semi-volatile VOCs, possibly introducing sampling artifacts (Zhao et al., 2013). As this was judged to be a greater drawback, an inlet filter was omitted. As such, both gas- and aerosol-phase VOCs are sampled; the reported concentrations represent the sum of these contributions. The presence of ozone in the sample cartridges may contribute to oxidation of the most reactive VOCs between collection and analysis. The use of an ozone filter may help to mitigate this effect. The effect of ozone filters on the samples is therefore being evaluated in ongoing work."

**Line 156: Please comment here on how atmospheric temperature was measured (see general comment 3)?**

Please see the response to general comment 3 above.

**Line 157: "It outputs analog voltage..." Is the same is true of the mass flow sensor, as well (i.e. produce an analog voltage that is converted into a flow value? Also, is this conversion based on laboratory or manufacturer based calibrations? Please comment in the text.**

Yes, the mass flow sensor also outputs an analog voltage. We have revised the text to clarify this. The conversion of the mass flow sensor is based on periodic laboratory calibrations. A sentence to this effect was included in the original version (see below). The revised text reads as follows:

"A mass flow sensor (Model D6F-P; Omron) was installed upstream of the pump to provide a continuous analog voltage output signal corresponding to the mass flow at standard temperature and pressure. The flow sensor supports a flow range of 0 to 1000 sccm and includes a built-in

**cyclone dust segregation system, which diverts particulates from the sensor element. The mass flow sensor was calibrated periodically against a reference standard in the lab."**

Because the flow sensor measures mass flow, not volume flow, the data from the pressure sensor is not used in the VOC mixing ratio calculation. We have clarified this in our response to General Comment 3, above, and in the revised text. The pressure sensor is therefore used only for diagnostic purposes (i.e., to determine whether the flow system is functioning properly). Hence, the factory calibration was deemed sufficient for conversion of the pressure sensor signal. The description of the pressure sensor was modified to reflect this:

"Pressure system. An absolute pressure transducer (MX4100AP; NXP) is positioned adjacent to the flow sensor in order to measure the pressure in the flow path. The measured pressure is used also used with atmospheric temperature to convert mass flow rate to volumetric flow rate as UAV altitude changes.as a diagnostic of proper operation of the flow system. The device operates across a pressure range of 20 to 105 kPa. It outputs an analog voltage signal recorded by the microcontroller that can be converted to a pressure value using a function provided by the manufacturer. Laboratory calibration of the pressure sensor is possible but was deemed unnecessary due to its purely diagnostic function."

**Line 162: Please comment on the inline, wetted of solenoid valves and their potential VOC emissions to which cartridge samples could be exposed. Could this influence the detection limit of this system, particularly with sensitive analyzers such as GC-ToF-MS?**

The solenoid valves and all other wetted parts of the sampling system are positioned downstream of the sorbent cartridges so that the sampled air does not contact any sampler surfaces prior to passing through the cartridge. Hence, any contamination due to the solenoid valves or other flow system materials would only occur diffusively, and would also appear in the field blanks. We have not observed any such signals in the blanks that have interfered with detection of the target compounds or affected the detection limit beyond the levels already noted for the blanks (ca. 10 pg, ca. 2.5 pptv). To address this comment in the manuscript, we have added a statement at the beginning of Section 2.2 Sampler Description stating that the sorbent cartridges are positioned at the input of the sampler flow path to minimize contamination:

"The adsorbent cartridges are positioned at the inlet of the flow path to ensure that the sample air does not come in contact with any flow path surfaces prior to sampling as it could lead to contamination or loss of analytes."

We have also slightly modified the text in Section 2.3 Sampling Methods where this issue is discussed. The modified text is as follows:

"The sorbent cartridges are mounted at the sampler inlet to ensure that the sample gas that passes through the cartridges has not contacted other surfaces in the flow system, thus preventing potential analyte losses or contamination from the flow system *tubing components*."

Line 170: Are there additional sensors to system pressure and system flow on the sampling platform? If not, please specifically list these two sensors.

No, the pressure and flow sensors are the only two. The suggested revision has been made.

**Line 172: "via the power distribution board" is awkward phrasing- consider rewording.**

As suggested, we have revised the text as follows:

"The sampling system is powered by the UAV batteries via the 18 VDC power output on the Matrice 600. The UAV power supplies two voltage regulators which provide 5 VDC output for the pump, pressure and flow sensors, and Arduino Uno, and valve driver boards, and 24 VDC output for the valve manifold."

**L182- L189: Discuss the benefits of be able to measure high molecular weight compounds (C9-C30) of this approach, compared with others.**

Two major classes of biogenic VOCs, monoterpenes (C10) and sesquiterpenes (C15), fall in the C9-C30 range. Hence, the higher molecular weight range is needed to capture these and other potential compounds of interest. The text has been amended as follows to make this point more clearly:

"Tenax TA is a relatively weak sorbent that collects components with volatility less than benzene (e.g.,  $>C_6$ ) including monoterpenes, C10, and sesquiterpenes, C15, whereas Carbograph 5TD shows strong sorbate affinity and captures low-molecular-weight VOCs with carbon number of C3 to C8 (Woolfenden, 2010a) including isoprene, C5. The combination of these sorbent materials enables sampling of VOCs with carbon number from C3 to C30, covering the expected range of atmospheric compounds from biogenic and anthropogenic sources (Goldstein and Galbally, 2007)."

**Line 204-207: Do you base your sample volume collection on prior measurements in different environments? Can this be adjusted easily in the field or between flights?**

The sample volume is determined from the detection limit of the adsorbent cartridges based on past studies and on the desired detection limit for the VOC mixing ratio. This is discussed in the introduction on lines 86-91 and as applied to determination of the sample volume for this study on lines 196-206. Both the flow rate and the sample time affect the sample volume; both are easily adjustable in the field. We have made the following revisions to the text to clarify these points:

*Lines 141-143: The volumetric flow of the pump is a function of the pressure drop across the inlet and outlet, and is controlled via a manually adjustable pinch valve (Model 44560; US Plastic Corp.) at the output of the flow system.*

Lines 196-206: The total sample volume depends upon the flow rate and sample collection time. Both of these parameters are easily adjusted in the field between flights. The flow is adjusted using the manual pinch valve downstream of the pump. The flight time is programmed in the flight algorithm executed by the Arduino Uno microcontroller. A constant low volumetric flow rate is required to allow for optimal sorbent-sorbate interaction and uptake onto the sorbent matrix. A target flow rate of 150 sccm was defined to maximize both VOC capture efficiency and sample volume (Woolfenden, 2010b;Markes International Ltd., 2014). Based on the relationship between sample volume and minimum detection limit reported by past studies (ca. 10 pg, Pankow et al., 2012), a minimum sampling volume of 1.5 L per adsorbent cartridge collected, corresponding to ca. 2.5 ppt VOC, is targeted. This results in 10 min of sampling time per cartridge. Two to three cartridge samples of this volume can be collected in a single flight while also carrying out take-off/landing and transits between sampling locations. The Arduino Uno microcontroller provides the operational flexibility to obtain smaller or larger sample volumes by utilizing either more tubes and shorter collection times or fewer tubes and longer collection times, respectively, during a single flight.

**Line 213-215: "not influence the results"- can you expand on this?**

This references a sentence regarding the introduction of sample artifacts due to transport and storage. The study protocol followed established methods that have been shown to have minimal artifacts due these factors, and we expect the same to be true in this case. After consideration, however, we have removed the phrase referenced by the reviewer, which cannot be proven. The sentence now reads:

"Under proper transport and storage, sample artifacts were have been shown to be minimal and did not influence the results (Pollmann et al., 2005)."

**Line 221-222: Are these internal standards injected prior to sample collection as well or simply prior to sample analysis? Please explain this in the text.**

The internal standards are injected prior to sample analysis. To clarify, the text has been amended as follows:

"Internal standards tetramethylethylene and decahydronaphtalene are injected into each sample after collection and prior to analysis."

Line 240: This is a good description of the uncertainty and the detection limit. This detection limit and uncertainty do not seem compatible with the "nearly 2 ppt" listed in the abstract. Are they? If so, please explain.

As noted above, the Abstract has been revised to reflect the 3 ppt detection limit, so it is now consistent with the values presented here.

**Line 242: Please 1) discuss the purpose of the CFD simulations and 2) the uncertainties in the SOLIDWORKS Flow simulations.**

1) Purpose of the CFD simulations:

We have added a more detailed explanation of the questions relevant to adsorbent cartridge sampling that we aimed to address with the simulations. The explanation is included in the Discussion (Line 282), rather than the Experimental (Line 242, cited by the reviewer) where it immediately precedes and contextualizes the results. The additional text reads as follows:

"The possible effects of air circulation created by the UAV multicopter rotors on the sampling was considered. The flow field is also a factor in determining the sampler placement. There were two main questions to be addressed. The first was to determine the time scale at which the air in the sampling region beneath the UAV is flushed. If the flushing time scale is significantly less than the sampling time, then, rather than being drawn from a stagnant pool, the sampled air can be taken as representative of the surrounding air. The second was to determine the spatial scale of the disturbance created by the rotors, in order to assess whether smoothing of concentration gradients by rotor-induced mixing is likely to influence the measured values. Unlike many realtime sensors, which have integration times on the order of a second, cartridge samples were collected over relatively long time periods (minutes). Over this time period, atmospheric mixing serves to average out gas concentration gradients at fine spatial scales (< a few m). Gradients at this scale would therefore not be resolved by cartridge samples, even when not collected from a UAV platform. If the spatial scale of mixing induced by the UAV is smaller than that of the atmosphere itself over the sampling period, the perturbation of fine spatial scale gradients by the UAV circulation will not significantly affect the measured concentrations. Hence, the second critical question to be addressed by the CFD simulations is whether the spatial scale of atmospheric mixing induced by the UAV rotors is larger than the spatial scale of atmospheric mixing over the sampling period. If it is not, then the mixing due to the UAV should have little effect on the cartridge samples."

**2) Uncertainties in SolidWorks Flow simulations**

Some possible contributions to the uncertainty of the flow simulations are the domain size, the grid spacing, the use of solid disks to simulate the rotors, and the landing gear position (down instead of retracted). The domain size of +/-1 m and grid resolution were chosen to capture the majority of the flow disturbance around the drone while also working within computational limitations. For the same reason, sensitivity studies of the effect of changing the domain size or grid spacing were not performed, so the uncertainties associated with variations in these parameters are unquantified.

The magnitudes of the pressure variations around the drone (+/-100 Pa, or +/- 0.10%) speed variations of ca. +/-0.2 m s-1 or ca. 2 to 25% of speeds of 1 to 12 m s-1. A 25% increase of the calculated speeds would suggest a similar increase in the spatial scale for the dissipation of the resulting disturbance. Hence, we estimate a range for the mixing scale of +/-5 to 7 m.

Other studies are consistent with the results of our simulations. Villa et al. (2016b) measured the velocity fields around a smaller (3.7 kg) hexacopter and found that the downwash largely dissipated within 3 m of the drone. Ventura Diaz and Yoon (2018) performed high resolution CFD simulations of several quadcopter UAVs. The resulting velocity fields (cf. their Figure 10) were qualitatively similar to those obtained in the current study, though the extent of the perturbations was only +/-1 m. Both studies investigated smaller UAVs than used here. A larger drone would be expected to have a larger mixing volume, consistent with the results of our simulations.

Overall, allowance for possible uncertainties does not change the conclusion that mixing due to the drone is likely less important than atmospheric mixing over the time period of the samples.

The following changes have been made to the text:

Section 2.5: "CFD simulations are carried out using SOLIDWORKS Flow Simulation (Ver. 2017 SP3.0) (Waltham, USA). Dimensions and an input geometric model of the UAV are obtained from the DJI company (DJI Downloads). A box with the dimensions and location of the sampler is added to the geometry file. The propellers are simulated by discs of the same diameter, and to simulate a hovering UAV a downward velocity of 11 m s-1 is imposed through each disc so that the lift produced by the motors balanced the system weight. The domain size was 2.4 m in width and 2.0 m in height, with the UAV centered horizontally and at 1.2 m vertically. An adaptive grid was used, such that the grid spacing is smaller where gradients are larger. Boundary conditions include atmospheric pressure far from the UAV, which is set to 1 atm. As the actual pressure during sampling may differ from this value, it is used only as a baseline for comparison. The results are optimized by performing iterations until the pressure difference between the last two iterations was within 2 Pa. Uncertainties in the CFD simulations could arise from the choice of domain size or grid resolution, which were limited by available computational resources, or assumptions such as the use of solid disks to model the rotors. In flight the legs are retracted to horizontal. The simulations do not account for possible changes to the circulation patterns due to the retraction of the landing gear, although this effect is expected to be minor minor relative to the volume of the disturbance created by the drone (cf., Section 3)."

Section 3 (Results and Discussion): "The magnitudes of the pressure variations around the UAV (+/-100 Pa, or +/- 0.10%) correspond to speed variations of ca. +/-0.2 m s-1 or ca. 2 to 25% of speeds of 1 to 12 m s-1. A 25% increase of the calculated speeds would suggest a similar increase in the spatial scale for the dissipation of the resulting disturbance. Hence, we estimate a range for the mixing scale of +/-5 to 7 m."

**Line 264: It would be worthwhile to discuss the influence of rotor-wash potentially on measurements and their differences at altitudes of 60 m, 75 m, and 100 m. Are these measurements representative of 60 +/- 5m?**

Here we address the question of the volume sampled by the drone as well as General Comment 1, above, which asks us to address "**The dilution due to rotor-wash, which is a problem for all instruments without an inlet that extends beyond the turbulence induced by the multi-rotor platform.**"

We agree with the reviewer that the volume represented by the measurement and the effect of the UAV on this volume is critical to interpretation of the results. In contrast to previous studies, this study does not aim to measure concentrations in a high-concentration plume emitted from a point source into low-concentration background air with fast time resolution. Instead, we aim to measure the average concentration from a horizontally varying non-point source over an integration time of several minutes. We therefore think of the effect of the drone circulation as 'mixing' of concentration gradients in the surrounding air, rather than 'dilution', which suggests loss of signal due to the introduction of background air into the sample. That is, there is spatial averaging of the air sample within the mixing volume of the drone, but the sample itself is also an average over the sampling time. The key question, as outlined in the 'Purpose of the CFD Simulations' above, is whether the mixing volume due to the drone is larger or smaller than the spatial scale due to atmospheric mixing of the air sampled over a 10 minute period. The revised discussion of the drone mixing volume in the manuscript is included below. We conclude that

the mixing volume extends approximately +/-5 to 7 m above and below the UAV but that this volume is small compared to the vertical scale of atmospheric mixing over the sampling time period. Please also see the responses to Reviewer 2 regarding bias in the sample altitude and the comparison of samples at different altitudes.

"Figure 4b shows the calculated air velocity distribution around the UAV. The simulation suggests that air enters the sampling region experiences roughly laminar downward flow from above the propellers, undergoes turbulent recirculation to the UAV sampling region, and then is ejected below the UAV. The simulation shows that the air flushing time in the sample region is fast (i.e., several seconds) compared to the timescale of VOC sampling (i.e., 5-10 min). The velocity disturbance due to the rotors extends approximately 5 m above and below the UAV. This is consistent with the CFD study by Ventura Diaz and Yoon (2018), which suggested that for their smaller quadcopter (1.2 kg), the sample represented an air parcel extending approximately 1 m above the UAV. As expected for a larger drone, the disturbed air volume derived from *Figure 4 is significantly larger than in their study. The flow patterns are remarkably similar* considering the simplifying assumptions and lower grid resolution used in this study (cf. Section 2.5), lending credence to the general flow features shown in Figure 4. In addition, the simulation shows that the air flushing time in the sample region is fast (i.e., several seconds) compared to the timescale of VOC sampling (i.e., 5-10 min). The magnitudes of the pressure variations around the UAV (+/-100 Pa, or +/- 0.10%) correspond to velocity variations of ca. +/-0.2 m/s or ca. 2 to 25% of velocities of 1 to 12 m/s. A 25% increase of the calculated velocities would suggest a similar increase in the spatial scale for the dissipation of the resulting disturbance. *Hence, we estimate a range for the mixing scale of* +/-5 *to* 7 *m*. *The simulations thus indicate* that the sampler performs representative real-time sampling of ambient VOC concentrations averaged across several ±5 to 7 meters around the UAV. For comparison, the spatial scale of atmospheric vertical mixing over the sampling period (10 min) can be estimated from the relationship  $z = \sqrt{2K\tau}$ , where K is the eddy diffusivity,  $\tau$  is the time period, and z is the vertical distance. Estimates of the eddy diffusivity within 10 m above a forest canopy are in the range of approximately 2 to 15 m2 s-1 during the day, though the values are uncertain and vary with local meteorology and canopy roughness (Bryan et al., 2012; Saylor, 2013; Freire et al., 2017). K then generally increases with altitude for several hundred meters above the canopy (Wyngaard and Brost, 1984; Saylor, 2013). Using the canopy-top values as a lower limit on the eddy diffusivity at the UAV height results in an estimated lower limit on the vertical mixing scale of ca. 50 to 150 *m*, substantially larger than that due to the UAV. A manuscript treating atmospheric mixing above the forest canopy more explicitly using a large eddy simulation (LES) method is currently underway. Nevertheless, this estimate suggests that mixing due to the UAV is expected to exert minimal influence on the measured VOC mixing ratios."

**Also note if these samples were taken on ascending vertical profiles or separate flights (related to general comment 1).**

The samples were collected on separate flights, as was stated in the original text (line 262):

"Three samples were collected in separate flights at heights of 60 m, 75 m, and 100 m relative to the ground level at the tower location."

**Line 267: Were cartridges at the tower collected using an identical cartridge sampling system, including a pressure sensor in the flow path and a mass flow sensor or only a pump? Please describe this in the text.**

No, the tower samples were collected using a hand-held motorized pump (Model 210-1002, SKC). As this is a constant volume pump, pressure and temperature are needed to calculate the total moles of sample air. For the tower samples reported here, temperature and pressure were not measured simultaneously. A temperature of 25 C and pressure of 1.0 atm were used in the calculation. Uncertainties in the temperature of  $\pm -5$  C ( $\pm -2\%$ ) and pressure of  $\pm -10\%$  were used to estimate the uncertainty in the mixing ratios. When combined with the other uncertainties, this gives an overall uncertainty of 23% in the tower measurements. Table 1 has been updated with the corrected mixing ratio values and uncertainties. The following changes to the text have also been made:

"For comparison, VOC collections were performed concurrently atop the MUSA Tower with a hand-held motorized pump (Model 210-1002, SKC). These samples were collected using a volumetric flow rate of 200 secm cm3 min-1 and sampling time of 20 min for a total sample volume of 2.0 L. Mixing ratios were calculated from Eq. 1 using a pressure of 1.00 atm and temperature of 25 °C (measurements of temperature and pressure were unavailable). Uncertainties in pressure of +/-10% and temperature of +/-5 C (+/-2%) were used to estimate an overall uncertainty of 23% for the tower samples."

**Line 285-290: Discuss in the text more explicitly what the impact is of deviations in pressure in the sampling region. How would this specifically impact the representativeness of cartridge measurements?**

As discussed earlier, the mass flow sensor inherently accounts for changes in sample pressure and temperature. Therefore, small deviations in the pressure of the sampling region should not affect the measured total mass of air sampled, the resulting VOC mixing ratio, or the representativeness of the measurements. To make this point in the text, the following sentence has been added at line 285:

"Because the mass flow sensor inherently accounts for changes in sample pressure and temperature, small deviations in the pressure of the sampling region should not affect the measured total mass of air sampled or the resulting VOC mixing ratio. This result also suggests that any possible effects of UAV pressure fields on any pressure sensitive sensor mounted in this area would be small."

**L346-347: This second half of this sentence is a bit confusing. Isn't pre-programed GPSbased operation already employed? Is the goal to integrate that seamlessly into the DJI flight software?**

The long-term goal is to control the sampler from the remote controller or flight-control app through the drone's signal output. For the first generation sampler described in the manuscript, however, the drone flight was controlled by pre-programmed GPS-based operation, but there was no communication between the drone or remote controller and the sampling box. The sampler was programmed to open and close the sample valves at pre-determined times after takeoff. The GPS control program was synchronized with it based on these elapsed times (with an added buffer). For example, if the drone flight time to the first sampling point was 2 minutes from takeoff, the first sample would be initiated 3 minutes into the flight and last for 10 minutes. To clarify this point in the manuscript, we have revised the text as follows:

"A major goal of ongoing development of the sampler is to enable operation control of sampler functions and collection of sampler data from the tablet-based drone control software, either manually or as part of a pre-programmed GPS-based flight trajectory algorithm. In the current version, the flight trajectory is programmed with the drone control software, whereas and sampler operation is controlled by a stand-alone program on the Arduino Uno microcontroller. The two programs are synchronized in time from initialization with a short time buffer so that the drone arrives at the sampling location 1 min prior to opening the valve. Both of these operational modes require In order to fully integrate these functions, real-time communication among the sampler, the UAV on-board computer, and the user control interface on the tablet is required. The Arduino Uno microcontroller is unable does not have the capability to communicate with the UAV on-board computer. To address this issue, an ongoing the next step in the development is the replacement the Arduino Uno microcontroller with a Raspberry Pi miniature computer."

**L356: How high were the winds on these days that operation of the solenoid, pump or sensors failed? How typical are winds this high?**

On the days the sampler failed, the wind speeds were around 5 m/s. Winds > 4 m/s for short periods are observed relatively frequently (40-50% of sampling days). The sampler, however, does not always fail under these conditions. The failure rate over 128 flights (including flights after the period reported in the manuscript) is about 2.5%. In addition, changes made to ruggedize electrical connections in the sampling box and frequent inspection of the electrical connections (before each flight) have largely addressed this issue. To incorporate these points, we have amended the text as follows:

"This capability can be important to alert the user to problems during flight, such as the failure of valves or the pump to be activated, as has occurred occasionally on windy days (5% of flights with winds >4 m/s) due to strong vibration. This failure mode has largely been eliminated by reinforcing the electrical connections and inspecting them before each flight."

**P22 (Figure 4): The M600 Pro is not typically flown (and I imagine samples aren't collected) with the legs down for landing. How is the flow in these simulations altered when the M600 legs are retracted, if at all? See general comment 1.**

All samples were collected with the landing gear retracted. The reviewer raises a good point that the circulation patterns around the drone could be somewhat different with the landing gear retracted than in landing position. The simulations were run with the landing gear down because they are in that position in the CAD files provided by the manufacturer. Unfortunately, for logistical reasons, it would be difficult to run new simulations with the landing gear retracted. The co-author who ran the simulations (J. Baptiste) is no longer at Harvard, where the original

simulations were run. In addition, for licensing reasons, we no longer have access to the software package that was used previously.

Hence, we will address the reviewer's concern using alternate approaches.

First, as was discussed above, the key question in the context of cartridge sampling is whether the drone creates atmospheric mixing on a spatial scale larger than the atmospheric mixing that takes place within the sampling period. The conclusion is that the spatial scale of the air sampled over a 10 minute period due to atmospheric mixing is larger than the ca. +/-5 m mixing scale of the drone. The position of the landing gear in the simulations becomes an issue if it changes the mixing scale enough to change the answer to this question.

The landing gear are composed of slender carbon fiber rods. As Figure 4 shows, air is drawn downward from above the drone through the rotors. It then recirculates upward in the region beneath the drone where the sampler is mounted. Based on the figure, the absence of pressure or velocity gradients in the immediate vicinity of the legs suggests that the presence of the legs does not significantly perturb this flow. We therefore conclude that the position of the landing gear is unlikely to significantly alter the mixing scale suggested by the simulations.

Further, we have added a reference to a paper by Villa et al., (2016b), who measured the velocity fields around a smaller hexacopter drone (3.7 kg vs. 9.6 kg + 1.0 kg payload in the current study). The velocity fields deduced from their measurements show overall flow patterns consistent with the simulation results shown in Figure 4, although the spatial scale of the disturbance would be larger for a larger and heavier drone.

We have added the following sentence acknowledging the possible effect of the landing gear position to the manuscript (Line 250):

"In flight the legs are retracted to horizontal. The shown simulations do not account for possible changes to the circulation patterns due to the retraction of the landing gear, although this effect is expect to be minor relative to the volume of the disturbance created by the drone (cf. Section 3)."

**P22 (Figure 4): Please add a vertical scale and horizontal scale on Fig. 4a and Fig. 4b.**

The figure has been revised to include vertical and horizontal scales. We have also added a figure caption (below), which was inadvertently omitted in the earlier version.

**"Figure 4.** (a) Vertical pressure distribution and (b) air velocity distribution around the UAV from the CFD simulation. Pressure difference between the UAV sampling area and the area under the propellers was simulated to be less than 100 Pa indicating a minimal effect of pressure on sampling. The air velocity was 1.65 m s-1 upward around UAV sampling region, suggesting a fast air flushing time underneath the sampling box."

**Reviewer 2:**

General Comments: This is a very well-written manuscript describing the development of a VOC sampler for autonomous, drone-based sampling. The motivation and relevant background is thoroughly but succinctly presented in the introduction, and the platform and results are clearly and generally well-described. I recommend publication of the manuscript, pending the authors: 1) add some context for what results should be expected for vertical distribution of VOCs in Table 1, so that the reader can better interpret the results presented here, and 2) more satisfactorily explore the vertical sampling bias introduced by rotors drawing air down from above (or gather comments from an additional reviewer with substantial experience with the fluid dynamics of drones). The CFD analysis is laudable, but does not conform to experience in working with large drones C1 with payloads, where vertical disruption of plumes extends greater than 5 m in many cases, and the paper cited to suggest < 1 m disruption is based on drone platforms that are substantially smaller.

We thank the reviewer for the thoughtful and helpful comments, which have led to substantial improvements to the manuscript. Responses to individual comments, including the two in the summary paragraph above, and the corresponding manuscript revisions are detailed below.

**Specific comments:**

111 – Noteworthy that the sampler was placed on the platform underneath the drone. Downwash and eddies present a significant challenge in sampling underneath drones (as you explore later), leading many to mount sensors on top of the drone, where flow is laminar, or to extend a sampling inlet outside the rotor influence. CFD simulations are a helpful place to start, but ultimately you can learn a lot by just flying your specific platform through a smoke plume. You'll notice straight, laminar flow lines on top that extend from several meters above (depending on system mass) and a mess of eddies underneath. Dave Barrett and Scott Hersey at Olin College of Engineering presented on this in collaboration with Aerodyne at AAAR and AGU in 2016 – check their materials for more clues. This eddy issue matters less for your application than for their 1-Hz instrument, since you are not after time-dependent (i.e. highly spatially resolved) data, but rather bulk VOC mass over an entire flight segment. But is nonetheless an important consideration. Explore options to mount on top, or to extend a sampling inlet to a point horizontally outside rotor influence.

We agree with the reviewer that there are potential drawbacks to mounting the sampler beneath the drone. There are also advantages. Likewise, there are advantages and disadvantages to mounting it on top. One particular disadvantage to top mounting is that we have observed that the temperatures at the top surface of the drone can get extremely hot, particularly during the dry season. This could have a particularly detrimental effect on adsorbent cartridges due to the higher volatility of VOCs at higher temperatures. As noted by the reviewer, the presence of eddies underneath the drone is less of an issue for our application, where samples are collected over a 10 minute period. After weighing these factors, we conclude that the choice to mount the

sampler beneath the drone is a reasonable one for this particular application. We have added a discussion of these issues to Section 3 of the text, as quoted below:

"There are both advantages and disadvantages to mounting the sampler either atop or beneath the UAV. The advantages of top mounting include faster time response and potentially higher spatial resolution due to laminar flow and less mixing. Some disadvantages are the potential for bias in some measurements, such as of particles, due to sampling from laminar flow rather than well mixed air, and the potential for more vertical bias due to the strong laminar downwash of air above the UAV. In addition, the temperatures at the top surface of the UAV have been observed to become extremely hot (ca. 40 °C), particularly during the dry season. This is particularly problematic for collecting VOCs on adsorbent cartridges, as the sampling efficiency may be reduced at elevated temperatures. On the other hand, the advantages to mounting beneath the UAV are that the sampler is protected from direct sunlight and therefore cooler. Also, the flow beneath the UAV is well mixed, which avoids flow effects such as a bias towards large particles. Disadvantages, such as mixing of concentration gradients and decreased time resolution, are most significant for sensors with fast time response. A study by Villa et al. (2016b), however, explored the differences in measured concentrations of a suite of trace gases from a point source when the sensors were mounted above, below, and in the horizontal plane of a hexacopter UAV. Their results show similar dilution of the plume measured above and below the UAV, suggesting that the air sampled on top of the drone does not necessarily experience less mixing. A sample inlet mounted such that it extends horizontally outside of the rotor wash was the least affected by the UAV flow fields and could be a good solution for fast sensors. The presence of eddies underneath the drone is less of an issue for our application, where samples are collected over a 10 minute period. Atmospheric mixing and temporal averaging will smooth out mixing ratio gradients over this time period, so mixing by drone-induced eddies should have little effect on the measurement. Since the disadvantage of overheating if the sampler is mounted on top of the UAV potentially outweighs the disadvantage of sampling from the turbulent flow underneath, the decision to mount the sampler beneath the UAV is a reasonable one for this particular application."

**240 – CFD simulation parameters are described, though it's not explicit at this point why you did CFD simulation (I can assume where you're headed). I suggest giving some sense of the need/purpose for this simulation before introducing it.**

We have added material to the introduction to discuss the need for the CFD simulations to understand the flow fields around the drone and their possible effects on the measurements and in the discussion to describe the specific aims of the simulations. These changes are described in more detail in the response to Reviewer 1, who made a similar comment.

258 – The drone was launched and recovered from a platform above the canopy, but one of the key motivations for the drone-based sampling platform is to avoid the need for platforms and to be able to access more remote sampling locations. Can you speak to the usability of this platform in the types of contexts that motivate the study (i.e. those with dense canopies and no platforms)?

There were several reasons for launching the drone from the tower in this study. The first was inexperience. Until we gained expertise in flying the drone with the sampler, we were most comfortable maintaining visual contact with the drone. Secondly, in many places (including the US) regulations for the use of UAVs require that the pilot maintain visual contact. This may change in the future as the use of drones becomes more widespread. For flights without visual contact, a camera would be useful for visualizing the position of the drone. In order to reduce the payload weight, no camera was mounted on the drone during sampling flights. This could be changed by adding a small camera at the expense of a few minutes of flight time or by using a second drone with a camera. In order to fly in an area with a dense canopy and no tower, it would be necessary to have at least a small clearing in which to take off and maneuver the drone up through the canopy. With additional experience and a camera for visualization, this should be possible in the future.

**The following text has been added to the discussion:**

"Current regulations in some locations, including the US, require that the operator maintain visual contact with the UAV. This was also deemed best practice in the current study as users gained experience and comfort with flight operations. Launching the UAV from a tower permitted the pilot to maintain visual contact during flight. As another approach, the UAV sampler has also been flown in locations with hills where it is possible to visualize the top of the canopy over an area of lower elevation from an area of higher elevation. In the future, as regulations permit, navigation from the ground to above the canopy should be possible and would allow sampling in more remote and densely forested regions. A clearing of sufficient size to allow the UAV to be navigated would be required. A camera to provide remote visualization, either on the same drone or on a second companion drone, would aid in navigation outside of the pilots visual range."

262 – Given the note above, and the high velocity of air flow down through the rotors of the drone, I am not convinced that 60 m actually represented 60 m. I should be clear that I see your exploration of this with CFD modeling, but your model results conflict with my experience seeing drones sample smoke plumes in the field. With a slightly larger drone (S900) and slightly heavier payload (2.5 kg), I consistently see rotors draw down air from several (>/= 5) meters above mounted instruments in buoyant plumes. Experience suggests to me that your vertical sampling bias is greater than the 1 m suggested in line 294. Further, the result suggesting 1 m vertical bias in air sampling based on rotor air flow in Diaz and Yoon (2018) is based on a significantly smaller drone with no payload. Your large drone with payload will, necessarily, exert a greater vertical impact on air flows than theirs. This comment comes with the caveat that I am basing them solely on experience and observations with quad copters, and no modeling or detailed analysis of my own. I recommend either a brief review of this section - especially as it relates to altitude-ofsample bias - by a reviewer with greater expertise in the fluid mechanics of multi-rotor aircraft, or an addition of language that outlines the potential for vertical sampling bias on the order of several meters.

We agree with the reviewer, both that the vertical mixing volume is larger than +/-1 m around the drone, and that there is likely a bias in the sampling height due to the downward motion of air induced by the drone. The perturbation volume question is addressed in more detail in the response to Reviewer 1's comment on Line 262. To address the question of vertical bias, we have added the following text to the Discussion:

"As noted above, the sampled air is drawn systematically from above the altitude of the UAV. It is therefore expected that the sampled air represents an altitude slightly higher than the flight altitude. Based on a mixing volume extending 5 - 7 m above the drone, a vertical bias of ca. -3 m altitude is inferred."

**278 – "Reasonable consistency" is subjective. Quantify, and compare with either sampling+measurement uncertainties or previously published variability in VOC concentrations with height above canopy (or both).**

We have replaced the sentence referenced by the reviewer with the following:

"Nevertheless, the results demonstrate reasonable consistency between samples collected by the UAV and on the tower, separated by 711 m. They also suggest that vertical concentration gradients can be assessed using this method. The results for all fall within the expected range of concentrations (e.g., ca. <1-10 ppb for isoprene) for the near-canopy environment over the Amazon rainforest based on previous observations (Alves et al., 2016; Harley et al., 2004)."

282 – CFD modeling appears. I applaud the authors for attempting to address rotor influence in sampling. Ultimately, as I stated above, I expect the below-drone air flow perturbations to be less important for your application of 10 min resolution samples. But the bias introduced in the vertical resolution is of concern and my experience tells me that for a drone your size, the vertical extent of air disruption is substantially greater than the 1 m suggested here, based on results from a much smaller drone platform with no payload. I am, unfortunately, not the right reviewer to critique your CFD model run, and suggest that an additional reviewer explore this.

We thank the reviewer for the helpful comment and agree with all points. The question of vertical bias in the sampling height is addressed in response to the previous comment by this reviewer on Line 262. The vertical extent of air disruption is discussed in response to Reviewer 1's comment on Line 262.

Table 1 – Can you put these results in context that help the reader understand the consistency of measurements and how they conform to expectation? For example, I notice that isoprene concentrations vary substantially with altitude, though not in a way that decays with altitude (as I might expect). Same with Pinene(s). As presented, I'm unable to discern why the 100 m sample at the sampling site has higher concentrations of monoterpenes than both the 60 m and 75 m sample. Can anything be determined from ratios of VOCs to tell what's going on here? What should I expect to see in vertical variability? This doesn't conform to my expectations of reducing concentration with

**altitude, so please explore this so that the reader isn't left with questions about whether sampling bias or the drone platform is responsible.**

For samples collected simultaneously at different altitudes above a single location, we would indeed expect a gradient of decreasing concentrations with height. Other variables can, however, influence concentrations in different locations, such as different canopy sub-types with different emission rates. VOC emissions also respond strongly to changes in light and temperature, so concentrations at a single location can vary strongly over periods of a few hours or even minutes. As a result, it is difficult to make direct comparisons between the samples presented in Table 1, which were all collected at different locations (tower vs. point A), altitudes, and times. For example, as mentioned, the 100 m sample at point A has a higher concentration than those collected at 60 and 75 m, but it was also collected closer to early afternoon (13:15 - 13:35 h), when VOC emissions typically peak, than were the 75 m (11:15 - 11:35 h) or 60 m (15:15 - 15:35 h) samples.

[revised manuscript text omitted]